# The RIX domain defines a class of polymorphic T6SS effectors and secreted adaptors

Katarzyna Kanarek[1], Chaya Mushka Fridman[1], Eran Bosis [2] ✉ & Dor Salomon [1] ✉

Bacteria use the type VI secretion system (T6SS) to deliver toxic effectors into bacterial or eukaryotic cells during interbacterial competition, host colonization, or when resisting predation. Identifying effectors is a challenging task, as they lack canonical secretion signals or universally conserved domains. Here, we identify a protein domain, RIX, that defines a class of polymorphic T6SS cargo effectors. RIX is widespread in the *Vibrionaceae* family and is located at N-termini of proteins containing diverse antibacterial and anti-eukaryotic toxic domains. We demonstrate that RIX-containing proteins are delivered via T6SS into neighboring cells and that RIX is necessary and sufficient for T6SS-mediated secretion. In addition, RIX-containing proteins can enable the T6SS-mediated delivery of other cargo effectors by a previously undescribed mechanism. The identification of RIX-containing proteins significantly enlarges the repertoire of known T6SS effectors, especially those with anti-eukaryotic activities. Furthermore, our findings also suggest that T6SSs may play an underappreciated role in the interactions between vibrios and eukaryotes.

The type VI secretion system (T6SS) is a protein delivery apparatus in Gram-negative bacteria, which was originally described as an anti-eukaryotic determinant[1,2]. Nevertheless, further investigations revealed that most T6SSs play a role in interbacterial competition[3,4], whereas only a few T6SSs have been identified as anti-eukaryotic[5,6]. These roles are mediated by toxic proteins, called effectors, which are deployed inside neighboring bacterial and eukaryotic cells[5,7,8].

Effectors are loaded onto a missile-like structure, which is ejected by a contractile sheath that engulfs it in the cytoplasm of the secreting cell[9,10]. The ejected apparatus is composed of Hcp proteins that are stacked as hexameric rings forming an inner tube; the tube is capped by a spike complex comprising a VgrG trimer sharpened by a PAAR repeat-containing protein (hereafter referred to as PAAR)[10]. T6SS effectors can be divided into two types: (1) specialized effectors, which are secreted structural components (i.e., Hcp, VgrG, or PAAR) containing a C-terminal toxic domain extension[11–14], and (2) cargo

effectors, which are toxic proteins that non-covalently interact with one of the missile components or its C-terminal extension[15–19], either directly or aided by an adaptor protein[20,21] or a co-effector[22]. Because cargo effectors can bind to diverse loading platforms on the tube or spike, they lack a canonical secretion signal or domain. Therefore, identifying cargo effectors is a challenging task, especially if they are not encoded within T6SS gene clusters or near T6SS-associated genes.

Many T6SS cargo effectors belong to one of three known classes of polymorphic toxins: (1) MIX (Marker for type sIX) domain-containing proteins[15,23]; (2) FIX (Found in type sIX) domain-containing proteins[16]; or (3) Rhs (Rearrangement hotspot) repeat-containing proteins[24–28]. These three domains are found N-terminal to diverse C-terminal toxic domains, and they are predicted to play a role in their delivery. MIX and FIX domains are specifically found in T6SS-secreted proteins, whereas Rhs repeats are also present in toxins secreted by other types of secretion systems. Notably, many Rhs

[1]Department of Clinical Microbiology and Immunology, Faculty of Medicine, Tel Aviv University, Tel Aviv, Israel. [2]Department of Biotechnology Engineering, Braude College of Engineering, Karmiel, Israel. ✉e-mail: bosis@braude.ac.il; dorsalomon@mail.tau.ac.il

repeats are fused to an N-terminal PAAR or PAAR-like domain and are therefore specialized effectors[27,29]. Nevertheless, many other cargo effectors lack a known domain at their N-terminus; it is possible that these effectors contain N-terminal delivery domains that have not yet been revealed.

We previously identified Tme1, an antibacterial T6SS effector of *Vibrio parahaemolyticus*. Tme1 contains a C-terminal toxic domain named Tme (T6SS membrane-disrupting effector), which permeabilizes membranes and dissipates membrane potential[30]; the sequence N-terminal to the Tme domain does not contain a known domain or activity. Here, we show that the N-terminus of Tme1 is necessary and sufficient to mediate T6SS secretion in *V. parahaemolyticus*, and we use it to reveal a new domain that is widespread in members of the *Vibrionaceae* family. Proteins containing the identified domain, named RIX (aRginine-rich type sIX), are secreted via T6SS. RIX is found N-terminal to diverse C-terminal extensions with antibacterial and anti-eukaryotic toxic activities, as well as to sequences that function as loading platforms for cargo effectors. Therefore, we reveal a new class

of T6SS-secreted proteins, including polymorphic effectors and secreted adaptors.

## Results

### The N-terminus of Tme1 is necessary and sufficient for T6SS-mediated secretion

We previously reported that Tme1 is an antibacterial effector delivered by T6SS1 in *V. parahaemolyticus* BB22OP[30]; its toxic domain, Tme, is located at the C-terminus. We hypothesized that the N-terminus of Tme1 plays a role in secretion via T6SS. To test this hypothesis, we monitored the expression and secretion of a truncated version of Tme1 lacking the first 60 amino acids (Tme1[61–310]) (Fig. 1a). As shown in Fig. 1b, truncating the 60 N-terminal amino acids abrogated Tme1 secretion, indicating that this region is necessary for T6SS-mediated secretion. Next, we sought to determine whether the N-terminus of Tme1 is sufficient for T6SS-mediated secretion. To this end, we used a non-T6SS protein, superfolder GFP (sfGFP). sfGFP alone was not secreted via *V. parahaemolyticus* T6SS1 (Fig. 1c, right lanes). However,

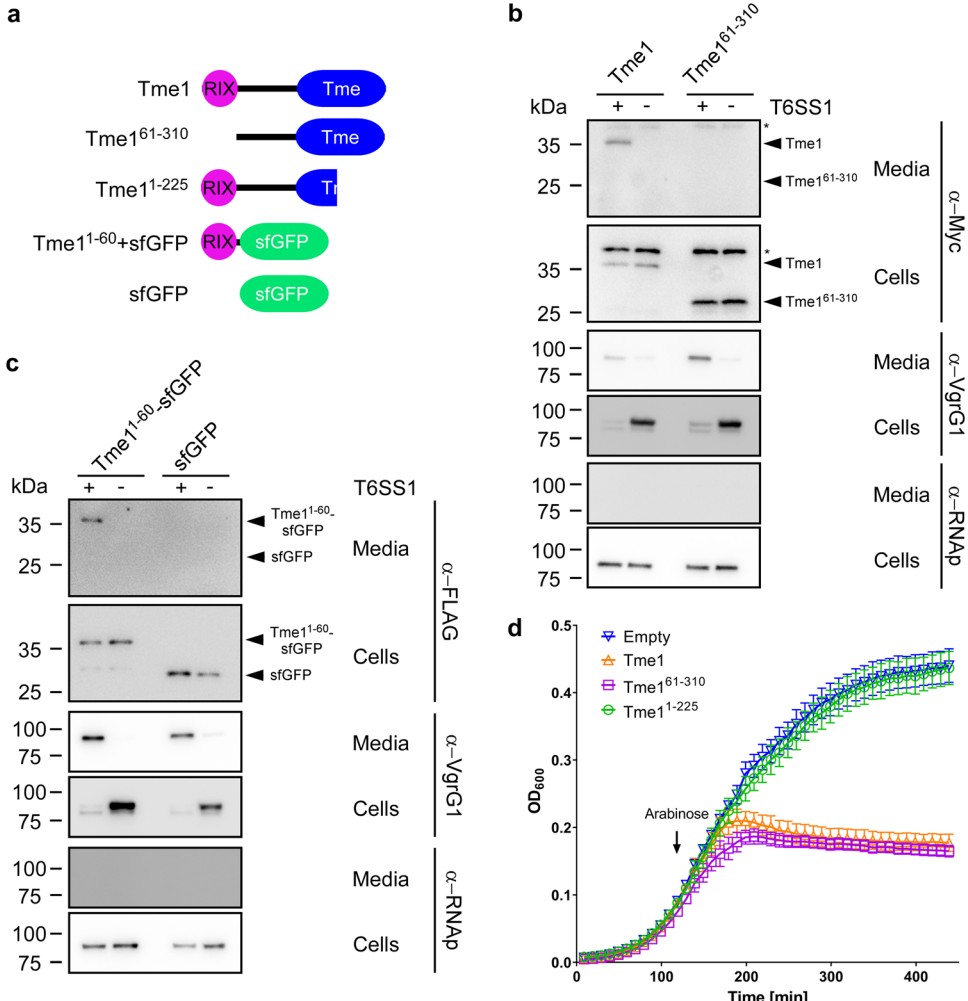

**Fig. 1 | The N-terminus of Tme1 is necessary and sufficient for T6SS-mediated secretion in *V. parahaemolyticus*. a** Schematic representation of Tme1 truncations and fusion proteins used in this figure. **b**, **c** Expression (cells) and secretion (media) of the indicated C-terminal Myc- and FLAG-tagged proteins expressed from pBAD33.1-based plasmids in *V. parahaemolyticus* BB22OP Δ*hns*/Δ*tme1* (T6SS1+; deletion of *hns* was used to hyper-activate T6SS1) and *V. parahaemolyticus* BB22OP Δ*hns*/Δ*tme1*/Δ*hcp1* (T6SS1−). Samples were grown in MLB supplemented with chloramphenicol and 0.01% (wt/vol) L-arabinose (to induce expression from plasmids) for 3 h at 30 °C. An asterisk denotes a non-specific band detected by the antibody. VgrG1 was used as a T6SS1 secretion positive control; RNA polymerase

sigma 70 (RNAp) was used as a loading and lysis control. The experiments were repeated three independent times (b) or two independent times (c) with similar results. Results from representative experiments are shown. **d** Growth of *E. coli* BL21 (DE3) containing pPER5 plasmids for the arabinose-inducible expression of the indicated proteins fused to an N-terminal PelB signal peptide (for delivery to the periplasm). An arrow denotes the timepoint (120 min) at which arabinose (0.05% wt/vol) was added to the media. Data are shown as the mean ± SD; *n* = 4 biological samples. The experiment was repeated four independent times with similar results. Results from a representative experiment are shown. Source data are provided as a Source Data file.

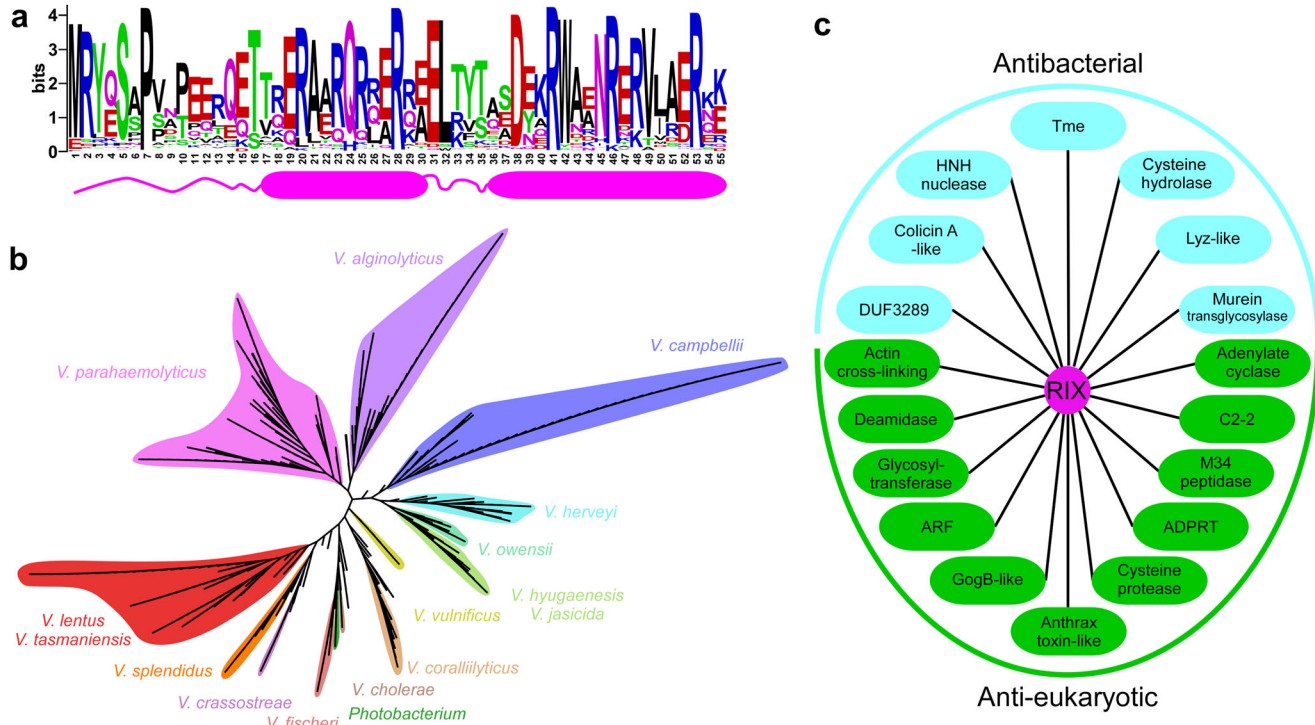

**Fig. 2 | RIX is found in polymorphic toxins that are widespread in vibrios. a** A conserved motif found at the N-terminus of polymorphic toxins (RIX) is illustrated using WebLogo, based on a multiple sequence alignment of sequences homologous to the N-terminal 55 amino acids of Tme1. The position numbers correspond to the amino acids in Tme1. A secondary structure prediction, based on the AlphaFold2 prediction of the Tme1 structure, is shown below. Alpha helices are denoted by cylinders. **b** Phylogenetic distribution of bacteria encoding a protein with a RIX domain, based on the DNA sequence of *rpoB* coding for DNA-directed RNA polymerase subunit beta. The evolutionary history was inferred using the neighbor-joining method. **c** Examples of known and predicted activities and domains found in sequences C-terminal to RIX domains. Lyz, lysozyme; ADPRT, ADP ribosyl transferase; ARF, ADP ribosylation factor.

fusing the N-terminal 60 amino acids of Tme1 to sfGFP (Tme1$^{1-60}$-sfGFP) enabled its secretion via *V. parahaemolyticus* T6SS1 (Fig. 1c, left lanes). Taken together, these results indicate that a region found at the N-terminus of Tme1 is necessary and sufficient for T6SS-mediated secretion.

To confirm that the N-terminus of Tme1 does not play a role in the effector's antibacterial activity, we monitored the toxicity of Tme1 variants with truncations at the N-terminus and the C-terminus. As shown in Fig. 1d, truncating the first 60 amino acids of Tme1 did not affect the toxicity of this effector in the periplasm of *E. coli*, whereas truncating the last 85 amino acids of Tme1, corresponding to the end of the Tme domain, abrogated its toxicity. The expression of all Tme1 forms was detected in immunoblots (Fig. S1). Thus, the N-terminal end of Tme1 is not required for toxicity.

### RIX is an arginine-rich domain found at N-termini of polymorphic toxins

Following these findings, we set out to identify regions homologous to the N-terminus of Tme1 in other proteins. We identified 502 unique protein accession numbers that contain a homologous sequence at their N-terminus (Supplementary Data 1). A multiple sequence alignment of these homologs revealed a conserved, arginine-rich motif corresponding to amino acids 1-55 in Tme1 (Fig. 2a); hereafter, we will refer to this region as the RIX (aRginine-rich type sIX) domain. RIX domain-containing proteins are encoded by 692 Gammaproteobacterial strains, exclusively belonging to the marine bacteria families *Vibrionaceae* (i.e., *Vibrio* and *Photobacterium*), and *Moritellaceae* (i.e., *Moritella*) (Fig. 2b and Supplementary Data 1). Many of the bacterial strains encoding RIX domain-containing proteins are pathogens of humans and animals; these include *V. parahaemolyticus*, *V. cholerae*, *V. vulnificus*, *V. campbellii*, *V. coralliilyticus*, and *V.*

*crassostreae*[31–34]. Importantly, although the identified RIX domain-containing proteins are not encoded within a T6SS gene cluster or auxiliary module, almost all (97.7%) of the genomes encoding RIX domain-containing proteins harbor a T6SS. Nevertheless, whereas most RIX domain-encoding strains harbor a T6SS that is similar to *V. parahaemolyticus* T6SS1, from which Tme1 is secreted, nearly 14% of the strains harbor a different T6SS (Supplementary Data 2). Notably, two RIX domain-containing proteins from *V. kanaloae* (WP_240696908.1 and WP_241908865.1) are fused to an N-terminal PAAR domain. These findings imply a genetic association between RIX and T6SS.

Analysis of the amino acid sequences C-terminal to RIX based on all-against-all pairwise similarity (using the CLANS classification tool[35]) revealed 33 distinct clusters (Table 1 and Fig. S2). Remarkably, the majority of these C-terminal sequences contain domains that are known or predicted to be toxins with anti-eukaryotic activities (e.g., actin cross-linking, deamidase, adenylate cyclase, and glycosyl-transferase) or antibacterial activities (e.g., pore-forming, HNH nuclease, and lysozyme-like) (Fig. 2c and Table 1). RIX domain-containing proteins with predicted antibacterial toxic domains are encoded upstream of a gene that possibly encodes a cognate immunity protein; genes encoding predicted anti-eukaryotic toxic domains do not neighbor a potential immunity gene. Notably, AlphaFold2 structure predictions[36,37] of representatives from each cluster revealed a possible conserved RIX structure, comprising two alpha helices preceded and connected by short loops (Fig. 2a and Fig. S3).

### RIX domain-containing proteins are secreted by T6SS

The results thus far indicated that: (1) Tme1 is a RIX domain-containing T6SS effector, (2) nearly all RIX domain-containing proteins are encoded by bacteria with a T6SS, and (3) most RIX domain-containing

**Table 1 | RIX domain-associated C-terminal extensions**

| Predicted role | Predicted C-terminal activity/domain[a,b,c,d] | Representative accession number | Number of unique proteins identified | An annotated putative immunity present[e] |
|---|---|---|---|---|
| Antibacterial | T6SS membrane-disrupting (Tme)[a] | WP_015297525.1 | 137 | Yes |
| | Colicin A-like pore-forming[b] | WP_130243383.1 | 42 | Yes |
| | HNH nuclease[d] | WP_130243482.1 | 24 | Yes |
| | Cysteine hydrolase[b] | WP_099165670.1 | 11 | No[*] |
| | Lyz-like[d] | WP_012127347.1 | 11 | Yes |
| | Unknown_1 | WP_169608408.1 | 6 | Yes |
| | DUF3289[d] | WP_152430150.1 | 4 | Yes |
| | Unknown_2 | WP_005534959.1 | 3 | Yes |
| | Murein Transglycosylase[b] | WP_042605467.1 | 1 | No[*] |
| Anti-eukaryotic | Glycosyltransferase_2[b] | WP_009697472.1 | 72 | No |
| | Deamidase_2[b] | WP_005530005.1 | 63 | No |
| | ADP ribosylation Factor (ARF)[c] | WP_006961831.1 | 36 | No |
| | ADP ribosyl transferase ADPRT_3[b] | WP_045402837.1 | 27 | No |
| | Glycosyltransferase_1[b] | WP_102362992.1 | 21 | No |
| | Deamidase_1[b] | WP_005536620.1 | 6 | No |
| | ADPRT_2[b] | WP_171802566.1 | 5 | No |
| | GogB-like[d] | WP_142566755.1 | 4 | No |
| | ADPRT_1[d] | WP_246210427.1 | 3 | No |
| | Actin cross-linking (ACD)[d] | WP_038863399.1 | 3 | No |
| | Unknown_3 | WP_012533558.1 | 3 | No |
| | Cysteine protease_1[c] | WP_186004041.1 | 1 | No |
| | Cysteine protease_2[c] | WP_137358098.1 | 1 | No |
| | Adenylate cyclase[b] | WP_146866449.1 | 1 | No |
| | Anthrax toxin LF-like[b,c] | WP_082041646.1 | 1 | No |
| | Tox-ART-HYE1[b] | WP_048666209.1 | 1 | No |
| | M34 peptidase[d] | WP_146866447.1 | 1 | No |
| | Glycosyltransferase_3[a,b] | WP_192890785.1 | 1 | No |
| | C2-2[b] | WP_236797446.1 | 1 | No |
| Secreted adaptor | TTR_1[c] | WP_157622110.1 | 4 | - |
| | Secreted adaptor for Tae4 | WP_095560627.1 | 3 | - |
| | TTR_2[c] | WP_152436132.1 | 2 | - |
| Unknown | Unknown_4 | WP_005537230.1 | 2 | No |
| Truncated | None | WP_107210948.1 | 1 | - |

[a]Predicted by homology to a known effector.
[b]Predicted using HHpred.
[c]Predicted using AlphaFold2 structure prediction followed by analysis in the Dali server.
[d]Predicted using NCBI's CDD.
[e]Assessed manually by inspecting the downstream encoded proteins.
[*]An unannotated open reading frame was manually identified.

proteins have predicted C-terminal toxic domains. Altogether, these results led us to hypothesize that RIX domain-containing proteins are a new class of polymorphic T6SS effectors. To test this hypothesis, we investigated whether a predicted RIX domain-containing protein from *V. campbellii* ATCC 25920, WP_005534959.1, which has a C-terminal domain of unknown function (Unknown_2; Table 1 and Fig. S4a) functions as a T6SS effector. We first monitored the toxicity of this protein in bacteria. Expression of WP_005534959.1 alone in *E. coli* was detrimental, whereas co-expression of the downstream-encoded protein (WP_005534960.1) antagonized this toxicity (Fig. S4b). We then tested the T6SS-mediated delivery of the predicted effector using a previously established T6SS surrogate system in a *V. parahaemolyticus* RIMD 2210633 derivative strain[30,38]. We found that expression of the predicted effector and immunity proteins from a plasmid in the surrogate attacker strain led to the killing of a parental prey strain lacking the predicted immunity protein (Fig. 3a). The killing was T6SS-dependent, since expression of the proteins in a derivative surrogate strain in which T6SS1 is inactive (Δ*hcp1*) did not lead to the killing of

the prey strain. Expression of the predicted immunity protein from a plasmid in the prey strain protected it from this attack. These results indicate that the RIX domain-containing WP_005534959.1 and its downstream encoded WP_005534960.1 are an antibacterial T6SS effector and immunity pair.

To further support our hypothesis, we next determined whether RIX domain-containing proteins belonging to diverse clusters are secreted by T6SS. To this end, we monitored the expression and secretion of three additional RIX-containing proteins encoded by *V. campbellii* ATCC 25920: WP_005536620.1, WP_005530005.1, and WP_038863399.1; these proteins are predicted to have anti-eukaryotic toxic domains at their C-terminus (Deamidase_1, Deamidase_2, and Actin cross-linking, respectively; see Table 1). Notably, the genomic neighborhood of their encoding genes did not include T6SS-associated genes which would suggest that these RIX domain-containing proteins are T6SS effectors (Fig. S5 and Supplementary Data 1). Remarkably, the three RIX domain-containing proteins were secreted from a surrogate strain in a T6SS-dependent manner, and

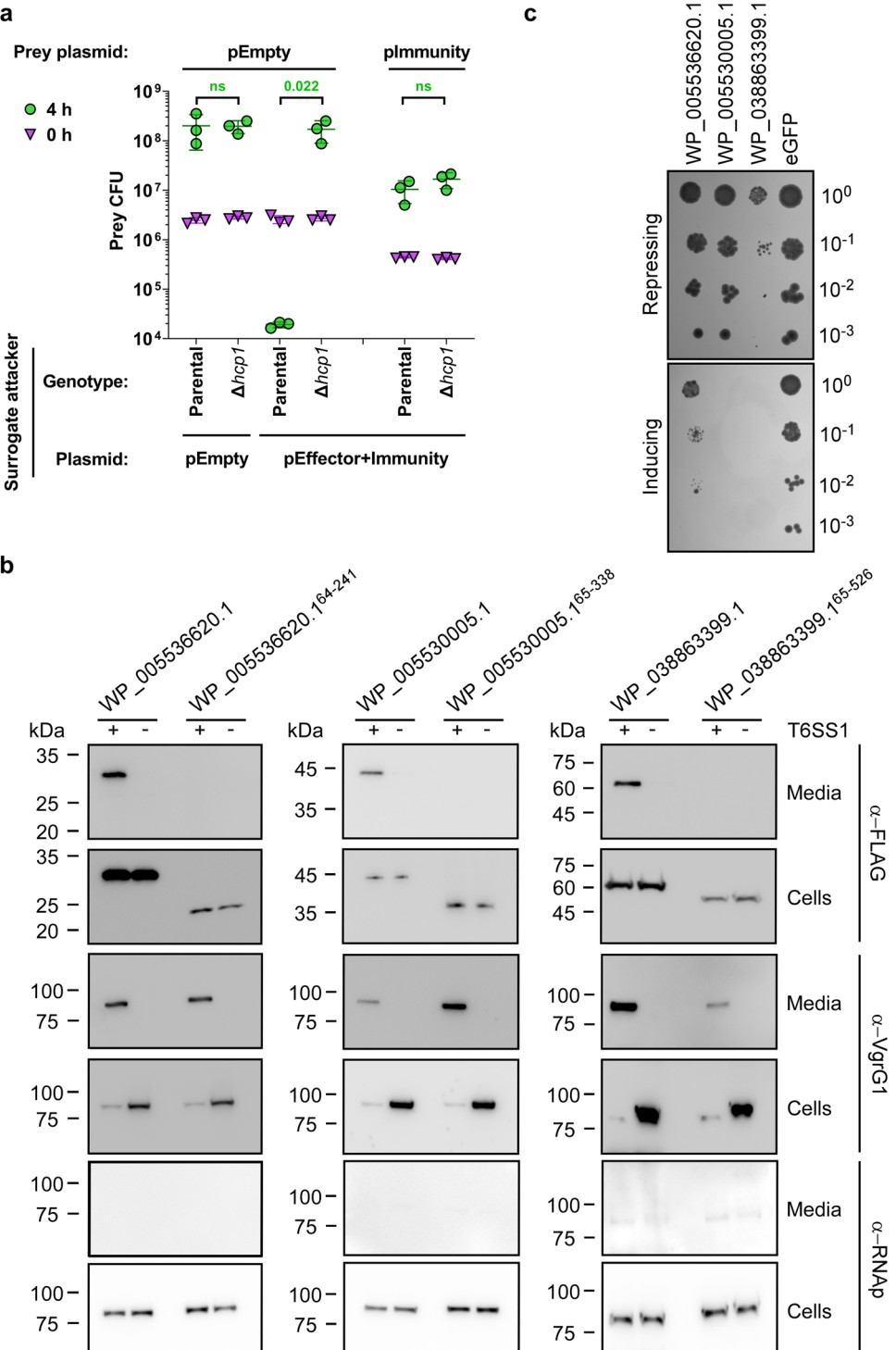

**Fig. 3 | RIX domain-containing proteins are delivered and secreted by T6SS.**
**a** Viability counts (CFU) of *V. parahaemolyticus* RIMD 2210633 Δ*hcp1* prey strains harboring either an empty plasmid (pEmpty) or a plasmid for the arabinose-inducible expression of WP_005534960.1 (pImmunity) before (0 h) and after (4 h) co-incubation with a surrogate attacker strain (*V. parahaemolyticus* RIMD 2210633 derivative) or its T6SS1⁻ derivative (Δ*hcp1*) carrying an empty plasmid or a plasmid for the arabinose-inducible expression of WP_005534959.1 and WP_005534960.1 (pEffector+Immunity). The statistical significance between samples at the 4 h timepoint was calculated using an unpaired, two-tailed Student's *t*-test; ns, no significant difference ($p > 0.05$). Data are shown as the mean ± SD; $n = 3$ biological samples. The experiment was repeated three independent times with similar results. Results from a representative experiment are shown. **b** Expression (cells) and secretion (media) of the indicated C-terminal FLAG-tagged proteins expressed

from pBAD33.1ᶠ-based plasmids in a *V. parahaemolyticus* RIMD 2210633-derivative surrogate (T6SS1⁺) strain or its Δ*hcp1* derivative (T6SS1⁻). Samples were grown in MLB supplemented with chloramphenicol and 0.01% (wt/vol) L-arabinose (to induce expression from plasmids) for 4 h at 30 °C. VgrG1 was used as a T6SS1 secretion positive control; RNA polymerase sigma 70 (RNAp) was used as a loading and lysis control. The experiment was repeated two independent times with similar results. Results from a representative experiment are shown. **c** Tenfold serial dilutions of yeast strains carrying pGML10 plasmids for the galactose-inducible expression of the indicated C-terminal Myc-tagged protein were spotted on repressing (4% glucose) or inducing (2% wt/vol galactose and 1% wt/vol raffinose) agar plates. eGFP, enhanced GFP. The experiment was repeated three independent times with similar results. Results from a representative experiment are shown. Source data are provided as a Source Data file.

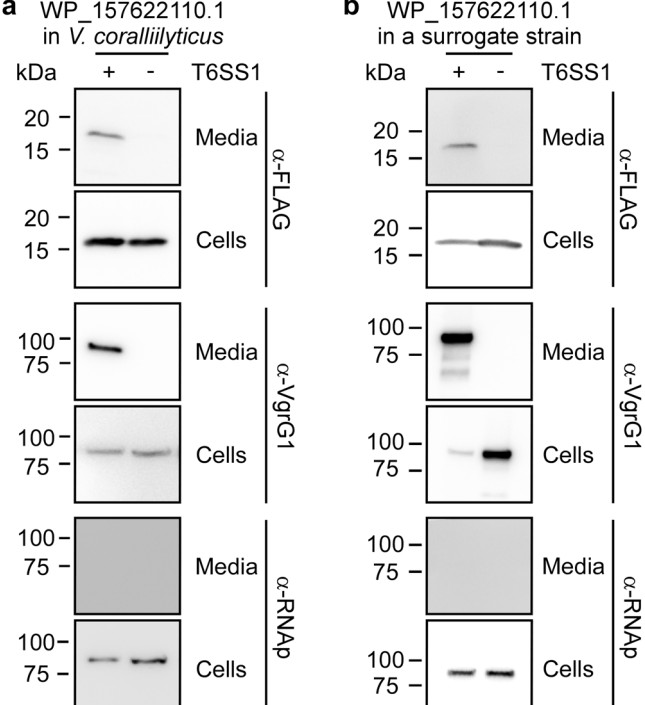

**Fig. 4 | RIX domain-containing proteins are secreted by their endogenous T6SS.** Expression (cells) and secretion (media) of a C-terminal FLAG-tagged WP_157622110.1 expressed from a pBAD33.1ᶠ-based plasmid in (**a**) a *V. coralliilyticus* BAA-450 wild-type strain (T6SS1⁺) or its Δ*hcp1* derivative (T6SS1⁻), or (**b**) a *V. parahaemolyticus* RIMD 2210633-derivative surrogate (T6SS1⁺) strain or its Δ*hcp1* derivative (T6SS1⁻). Samples were grown in MLB supplemented with chloramphenicol and 0.05% (wt/vol) L-arabinose (to induce expression from the plasmid) for 4 h at 30 °C. VgrG1 was used as a T6SS1 secretion positive control; RNA polymerase sigma 70 (RNAp) was used as a loading and lysis control. The experiments were repeated three independent times with similar results. Results from representative experiments are shown. Source data are provided as a Source Data file.

their secretion was abrogated in the absence of the RIX domain (Fig. 3b). Furthermore, they were all toxic when expressed in the eukaryotic yeast model organism, *Saccharomyces cerevisiae* (Fig. 3c and Fig. S6a), suggesting that they affect a conserved eukaryotic target[39]. Indeed, we found an actin ladder indicative of actin cross-linking[13] in the lysates of yeast expressing the RIX domain-containing protein with the C-terminal actin cross-linking domain (Fig. S6b). Taken together, these results support our hypothesis that RIX domain-containing proteins are secreted by T6SSs.

Next, we sought to demonstrate the secretion of another RIX domain-containing protein via its endogenous T6SS, in addition to Tme1, which was shown previously (Fig. 1b). To this end, we investigated WP_157622110.1, encoded by *V. coralliilyticus* BAA-450. To monitor the secretion of WP_157622110.1 via T6SS in *V. coralliilyticus*, we set out to identify the conditions under which T6SS1 of *V. coralliilyticus* BAA-450 is active. First, we monitored the secretion of the T6SS spike protein VgrG1, and the ability of this strain to intoxicate *V. natriegens* prey bacteria during competition under different temperatures in media containing 3% (w/v) NaCl. Our results revealed that *V. coralliilyticus* BAA-450 T6SS1 is an antibacterial system that is active at 30 °C (i.e., under warm, marine-like conditions) (Fig. S7). We then monitored the secretion of the RIX domain-containing protein, WP_157622110.1, when expressed from a plasmid. As shown in Fig. 4a, WP_157622110.1 was secreted by *V. coralliilyticus* BAA-450 in a T6SS1-dependent manner, confirming the secretion of RIX domain-containing proteins via their endogenous T6SS. In addition, we

confirmed that this RIX domain-containing protein is secreted in a T6SS-dependent manner from a surrogate strain (Fig. 4b).

## Secreted RIX-containing proteins can enable the delivery of other cargo effectors

Unlike most RIX domain-containing proteins, which have a predicted C-terminal toxic domain, WP_157622110.1 has a predicted C-terminal TTR (Transthyretin-like) domain (TTR_1; see Table 1). This domain was previously identified at C-terminal extensions of secreted T6SS spike components, such as VgrG and PAAR, and it was suggested to function as an internal adaptor for cargo effectors[12,40-42]. Therefore, we reasoned that WP_157622110.1 is not a toxic effector.

In *V. coralliilyticus* BAA-450, WP_157622110.1 is encoded by the first gene in a three-gene operon; notably, homologous modules comprising the TTR domain and the two downstream encoded proteins are also found fused to the T6SS-secreted components PAAR and VgrG in other bacteria (Fig. 5a). The genetic composition of the three-gene operon and the presence of the TTR domain in WP_157622110.1 led us to hypothesize that: (1) the second and third genes in the operon encode an antibacterial T6SS cargo effector and its cognate immunity protein, and (2) the RIX domain-containing protein acts as a secreted adaptor that enables the effector's delivery via T6SS, possibly by tethering it to the T6SS.

Before addressing these hypotheses experimentally, we used AlphaFold2 to predict whether the RIX domain-containing protein interacts with its downstream encoded protein. Indeed, AlphaFold2 predicted that the C-terminus of the RIX domain-containing WP_157622110.1, corresponding to the TTR domain, interacts with the N-terminus of the predicted effector (WP_006958655.1; hereafter named Rte1 for RIX-tethered effector 1) to complete a barrel-like fold (Fig. 5b). This inter-domain prediction was made with a high degree of certainty, according to the AlphaFold2 predicted aligned error analysis (Fig. 5c). In addition, an AlphaFold2 structure prediction indicated that Rte1 and its predicted immunity protein, WP_050778602.1 (hereafter named Rti1 for RIX-tethered immunity 1), interact with a high degree of certainty (Fig. S8a, b). Furthermore, sequence (HHpred[43]) and structure prediction (AlphaFold2 prediction followed by Dali server analysis[44]) analyses suggested that Rte1 contains an ADPRT (ADP ribosyl transferase)-like domain with a fold similar to that of the pertussis toxin (Fig. S8a), to which Rti1 is expected to bind (Fig. S8a, b), and a C-terminal glycine-zipper-like domain. A conservation logo assembled from a multiple sequence alignment of Rte1 homologs revealed conserved residues (H94, S106, and E174) that are similar to the catalytic residues of ADPRT toxins[45], which localize to a cleft within the predicted Rte1 structure (Fig. S8a, c); this cleft corresponds to the ADPRT-like domain active site and is predicted to be occluded by Rti1 (Fig. S8a).

Using these predictions, we set out to test our hypotheses. First, we determined whether Rte1 is toxic when expressed in bacteria. Indeed, its expression in *E. coli* was toxic; however, it was antagonized by co-expressing Rti1 (Fig. S8d). Substitution of either H94 or S106, corresponding to conserved residues that were identified as the possible active site in the sequence and the structural analyses of Rte1, for alanine abolished the toxic activity of Rte1 in *E. coli* (Fig. S8e). The expression of all Rte1 forms in *E. coli* was confirmed in immunoblots (Fig. S8f).

Next, we tested whether Rte1 is secreted in a T6SS-dependent manner from the T6SS surrogate system, in the presence and absence of its cognate RIX domain-containing protein. To avoid self-intoxication of the surrogate strain by the effector, we used an inactive mutant of Rte1 (Rte1ᴴ⁹⁴ᴬ; see Fig. S8e). As shown in Fig. 5d, a plasmid-encoded Rte1ᴴ⁹⁴ᴬ was secreted in a T6SS-dependent manner when the upstream-encoded RIX domain-containing protein, WP_157622110.1, was co-expressed; however, in the absence of the RIX-domain-containing protein, Rte1ᴴ⁹⁴ᴬ was not detected in the medium.

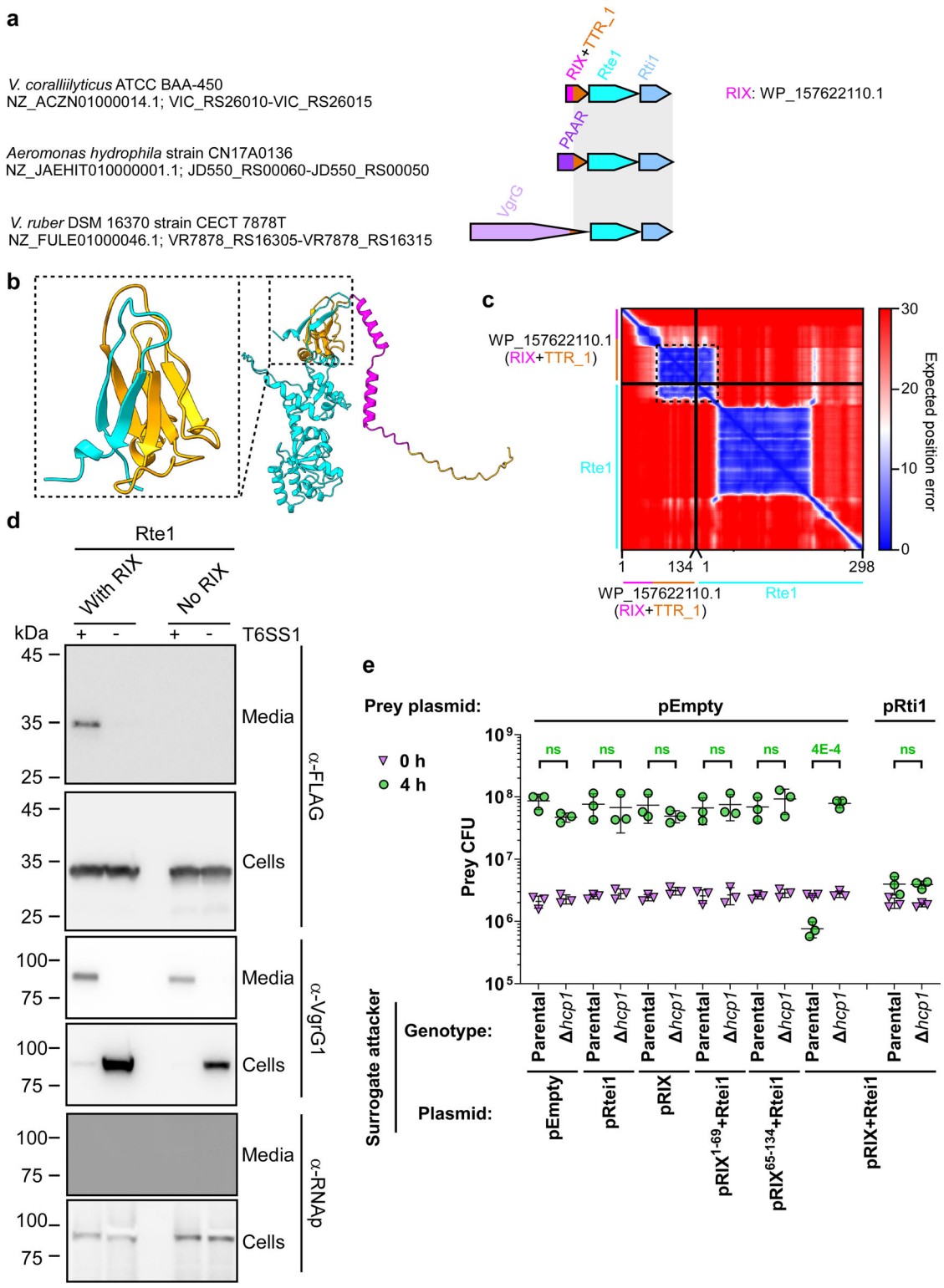

These results indicate that the RIX and TTR domains-containing protein is required for T6SS-mediated secretion of Rte1.

In addition, we investigated whether Rte1 and Rti1 function as antibacterial T6SS effector and immunity pair in which the effector is dependent on the RIX domain-containing protein for delivery. To this end, we performed self-competition assays using the surrogate T6SS system. As shown in Fig. 5e, T6SS-dependent toxicity against a sensitive prey strain was only observed when Rte1 and Rti1 were expressed together with the full-length, upstream-encoded RIX domain-containing protein (pRIX+Rtei1), but not when they were expressed alone (pRtei1) or with either a C- or an N-terminal truncation of the RIX domain-containing protein (pRIX$^{1-69}$+Rtei1 and pRIX$^{65-134}$+Rtei1, respectively). Notably, expression of the RIX domain-containing protein alone (pRIX) did not result in prey intoxication. Expression of Rti1 from a plasmid in the prey strain protected it from the T6SS-mediated toxicity. Taken together, our results support a role for a RIX domain-containing protein as a secreted adaptor that enables the delivery of a cargo effector via T6SS.

When we examined other RIX domain-containing proteins that had no predicted toxic domain at their C-terminus, we identified two

**Fig. 5 | RIX domain-containing proteins can enable the T6SS-mediated delivery of cargo effectors. a** The gene structure of the *V. coralliilyticus* BAA-450 operon encoding WP_157622110.1. Operons encoding a module homologous to the C-terminal extension of WP_157622110.1, Rte1, and Rti1 are shown below; a gray rectangle denotes the region of homology. Strain names, GenBank accession numbers, and locus tags are provided. Names of encoded proteins or domains are denoted above. TTR, Transthyretin-like. **b** An AlphaFold2 structure prediction of the complex between WP_157622110.1 (orange; RIX denoted in magenta) and Rte1 (cyan). The interaction interface, corresponding to amino acids 73-134 of WP_157622110.1 (orange) and 1-25 of Rte1 (cyan), is shown inside the dashed rectangle. **c** The predicted aligned error of the complex shown in (**b**). Low values indicate well-defined predicted relative position and orientation of two residues. **d** Expression (cells) and secretion (media) of the C-terminal FLAG-tagged Rte1$^{H94A}$ encoded alone (No RIX) or with WP_157622110.1 (With RIX), expressed from pBAD33.1$^F$-based plasmids in a *V. parahaemolyticus* RIMD 2210633-derivative surrogate strain (T6SS1$^+$) or its Δ*hcp1* derivative strain (T6SS1$^-$). Samples were grown in MLB supplemented with chloramphenicol and 0.01% (wt/vol) L-arabinose (to induce expression from plasmids) for 4 h at 30 °C. VgrG1 was used as a

T6SS1 secretion positive control; RNA polymerase sigma 70 (RNAp) was used as a loading and lysis control. The experiment was repeated three independent times with similar results. Results from a representative experiment are shown. **e** Viability counts (CFU) of *V. parahaemolyticus* RIMD 2210633 Δ*hcp1* prey strains harboring an empty plasmid (pEmpty) or a plasmid for the arabinose-inducible expression of Rti1 (pRti1) before (0 h) and after (4 h) co-incubation with a surrogate attacker *V. parahaemolyticus* RIMD 2210633 derivative or its T6SS1$^-$ derivative (Δ*hcp1*) carrying an empty plasmid or a plasmid for the arabinose-inducible expression of WP_157622110.1 (pRIX), Rte1 and Rti1 (pRtei1), or the operon including WP_157622110.1, Rte1 and Rti1, either in its wild-type form (pRIX-Rtei1), with a stop codon replacing the codon for amino acid 70 in WP_157622110.1 (pRIX$^{1-69}$-Rtei1), or with WP_157622110.1 starting at amino acid 65 (pRIX$^{65-134}$-Rtei1). The statistical significance between samples at the 4 h timepoint was calculated using an unpaired, two-tailed Student's *t*-test; ns, no significant difference ($p > 0.05$). Data are shown as the mean ± SD; $n = 3$ biological samples. The experiment was repeated three independent times with similar results. Results from a representative experiment are shown. Source data are provided as a Source Data file.

additional types of predicted RIX domain-containing secreted adaptors. As exemplified by WP_095560627.1 and WP_152436132.1, these RIX domain-containing proteins are encoded by the first gene in a three-gene operon (Fig. S9a). Notably, the effectors encoded by the second gene within these two operons are known: a Tae4 peptidoglycan amidase effector[17] and a homolog of the TseH cysteine hydrolase effector[42,46], respectively. The third gene in these operons encodes the predicted cognate immunity protein. Similar to the Rte1-containing module described above, a module encompassing the C-terminal extension of the RIX domain (i.e., the predicted loading platform), the effector, and the immunity protein can be found fused to a PAAR domain in other vibrios. Structure predictions using AlphaFold2 suggest that the RIX-fused C-terminal domain and the N-terminus of the downstream encoded effector interact (Fig. S9b), thus further supporting the notion that RIX domain-containing proteins can serve as secreted adaptors that enable the T6SS-mediated delivery of cargo effectors.

## Discussion

In this work, we identified a new class of T6SS-secreted proteins that are widespread in *Vibrionaceae*. These proteins share an N-terminal domain that we named RIX; however, they have polymorphic C-terminal extensions with predicted antibacterial, anti-eukaryotic, or cargo binding activities. We showed that RIX is necessary and sufficient for T6SS-mediated secretion, and we experimentally confirmed the T6SS-mediated secretion of six representative RIX domain-containing proteins: two antibacterial effectors (Tme1 and WP_005534959.1), three anti-eukaryotic effectors (WP_005536620.1, WP_005530005.1, and WP_038863399.1), and one secreted adaptor (WP_157622110.1). In addition, we identified a novel, non-RIX domain-containing T6SS cargo effector, Rte1. Notably, without RIX, the identification of these polymorphic T6SS effectors would not have been trivial, since RIX domain-encoding genes are orphan (i.e., they are not located within an operon containing other T6SS-associated genes), and because many of the predicted toxic domains found in RIX domain-containing proteins have not been previously associated with T6SS effectors.

Revealing many RIX domain-containing proteins that have predicted anti-eukaryotic toxic domains suggests that we have underappreciated the potential role played by T6SS in the interactions between vibrios and eukaryotic organisms, whether they are hosts or predators. Although T6SS was originally described as an anti-eukaryotic determinant, allowing *V. cholerae* to escape predation by grazing amoeba[1], most *Vibrio* T6SSs investigated to date were shown to play a role in interbacterial competitions[47–53], and only a handful of *Vibrio* T6SSs have been implicated in anti-eukaryotic activities. These *Vibrio* T6SSs include T6SS1 and T6SS3 in *V. proteolyticus*[49,54,55], T6SS1 in

*V. tasmanienses*[56], and a T6SS in *V. crassostreae*[57]. Several additional *Vibrio* T6SSs were also suggested to play a role in interactions with eukaryotes based on the presence of a predicted anti-eukaryotic MIX-effector in their genome[23]. Furthermore, only a few T6SS effectors with anti-eukaryotic activities have been described to date, in vibrios and in other bacteria[5]. Therefore, our identification of diverse families of RIX domain-containing effectors with known or predicted anti-eukaryotic activities substantially enlarges this list. We hypothesize that these effectors are mostly used by vibrios to defend against predation by protists, as previously shown for *V. cholerae*[1,58], rather than for colonization of aquatic or land animals. However, further work is required to identify the eukaryotic organisms that are naturally targeted by these potentially anti-eukaryotic RIX domain-containing effectors and to determine whether the role of these effectors is offensive or defensive.

Our finding that RIX domain-containing proteins are mostly restricted to vibrios reveals a previously unknown concept of family-specific T6SS secretion domains. Previously described domains that are required for T6SS secretion, such as MIX[15] and FIX[16], are widespread across different Gram-negative bacterial families. The mechanism governing such a restricted distribution of the RIX domain is unclear. Since RIX domain-encoding genes are orphans, it is likely that they are acquired via horizontal gene transfer and not together with the T6SS apparatus. It is possible that the transfer mechanism requires a component that is found only in vibrios. Alternatively, if RIX domain-containing proteins are secreted only by T6SS types that are themselves restricted to vibrios, then the distribution of these T6SSs might provide the evolutionary drive for maintaining or discarding acquired RIX domain-encoding genes. It is intriguing to speculate that additional family-specific T6SS secretion domains are found in *Vibrionaceae* or in other bacterial families.

In addition to effectors, RIX domains are also found in secreted T6SS adaptors, revealing a new mechanism of secreted adaptor-mediated delivery of T6SS cargo effectors. Our results indicated that RIX domains replace PAAR and VgrG proteins, which are secreted structural components of the T6SS, in modules comprising a C-terminal loading platform extension and a cargo effector and immunity pair. We experimentally demonstrated that a new cargo effector, Rte1, requires an upstream-encoded RIX domain-containing protein in order to be secreted by the T6SS and delivered into prey cells during bacterial competition. This secretion mechanism is similar to the recently described mechanism involving a MIX domain-containing co-effector[22], yet it is distinct since the RIX domain-containing protein is secreted via T6SS on its own, whereas the MIX-containing co-effector is dependent on its effector partner for secretion.

Notably, our attempts to biochemically identify the T6SS tube-spike component with which RIX domain-containing proteins interact have been inconclusive. Therefore, the identity of the loading platform for RIX domain-containing proteins on the T6SS remains an open question. Nevertheless, we speculate that RIX-containing proteins are loaded onto the spike rather than inside the narrow Hcp tube. This hypothesis is supported by previous observations that large folded proteins are not secreted by T6SS[59], together with our findings that RIX was able to drive the secretion of the folding-enhanced sfGFP[60] and that a 526 amino acid-long RIX domain-containing protein is secreted via the T6SS. It is possible that the loading of RIX domain-containing proteins onto the ejected apparatus requires a yet-unknown component or adaptor, or that a fully assembled apparatus is required for proper effector loading. Therefore, further work is needed to determine the mechanism governing the secretion of RIX domain-containing proteins.

In conclusion, we identified a new class of polymorphic T6SS cargo effectors widespread in vibrios, and we revealed a new, secreted adaptor-mediated delivery mechanism for T6SS cargo effectors. Our findings considerably enlarge the pool of predicted anti-eukaryotic T6SS effectors, suggesting that the role played by *Vibrio* T6SSs in interactions with eukaryotes has been underappreciated. In future work, we will determine how RIX domain-containing proteins are loaded onto the secreted T6SS, and we will explore the potential use of RIX domains and RIX domain-containing secreted adaptors to deliver engineered, non-canonical cargo effectors via T6SS to be used as biotreatments[61] or molecular biology tools[62].

# Methods
## Strains and media
For a complete list of strains used in this study, see Table S1. *Escherichia coli* strains BL21 (DE3) and DH5α (λ-pir) were grown in 2xYT broth (1.6% wt/vol tryptone, 1% wt/vol yeast extract, and 0.5% wt/vol NaCl) or on Lysogeny broth agar (LB; 1.5% wt/vol) plates at 37 °C, or at 30 °C when harboring effector expression plasmids. The media were supplemented with chloramphenicol (10 µg/ml) and/or kanamycin (30 µg/ml) to maintain plasmids and with 0.4% (wt/vol) glucose to repress protein expression from the arabinose-inducible promoter, P*bad*. To induce expression from P*bad*, L-arabinose was added to the media at 0.05-0.1% (wt/vol), as indicated.

*Vibrio parahaemolyticus* strains BB22OP, RIMD 2210633, and their derivatives, as well as *Vibrio natriegens* ATCC 14048 and *Vibrio coralliilyticus* ATCC BAA-450, were grown in Marine Lysogeny broth (MLB; LB containing 3% wt/vol NaCl) and on Marine Minimal Media (MMM) agar plates (1.5% wt/vol agar, 2% wt/vol NaCl, 0.4% wt/vol galactose, 5 mM MgSO$_4$, 7 mM K$_2$SO$_4$, 77 mM K$_2$HPO$_4$, 35 mM KH$_2$PO$_4$, and 2 mM NH$_4$Cl) at 30 °C. The media were supplemented with chloramphenicol (10 µg/ml) or kanamycin (250 µg/ml) to maintain plasmids. To induce expression from P*bad*, L-arabinose was added to media at 0.01-0.1% (wt/vol), as indicated.

*Saccharomyces cerevisiae* BY4741 (MATa, his3Δ0, leu2Δ0, met15Δ0, and ura3Δ0) yeast were grown in Yeast Extract–Peptone–Dextrose broth (YPD; 1% wt/vol yeast extract, 2% wt/vol peptone, and 2% wt/vol glucose) or on YPD agar (2% wt/vol) plates at 30 °C. Yeast containing plasmids that provide prototrophy to leucine were grown in Synthetic Dropout media (SD; 6.7 g/l yeast nitrogen base without amino acids, 1.4 g/l yeast synthetic dropout medium supplement) supplemented with histidine (2 ml/l from a 1% wt/vol stock solution), tryptophan (2 ml/l from 1% wt/vol stock solution), uracil (10 ml/l from a 0.2% wt/vol stock solution), and glucose (4% wt/vol). For galactose-inducible expression from a plasmid, cells were grown in SD media or on SD agar plates supplemented with galactose (2% wt/vol) and raffinose (1% wt/vol).

## Plasmid construction
For a complete list of plasmids used in this study, see Table S2. For a complete list of primers used in this study, see Table S3. For expression in bacteria or yeast, the coding sequences (CDS) of the following protein accession numbers: WP_015297525.1 (Tme1), sfGFP, WP_005534960.1, WP_005536620.1, WP_005530005.1, WP_038863399.1, WP_157622110.1, WP_006958655.1, and WP_050778602.1 were PCR amplified from the respective genomic DNA of the encoding bacterium or from a plasmid. Next, amplicons were inserted into the multiple cloning site (MCS) of pBAD$^K$/Myc-His, pBAD33.1$^F$, or their derivatives using the Gibson assembly method[63]. Plasmids were introduced into *E. coli* BL21 (DE3) or DH5α (λ-pir) by electroporation, and into vibrios via conjugation. Transconjugants were selected on MMM agar plates supplemented with the appropriate antibiotics to select clones containing the desired plasmids. For galactose-inducible expression in yeast, genes were inserted into the MCS of the shuttle vector pGML10 (Riken) using the Gibson assembly method, in-frame with a C-terminal Myc-tag. Yeast transformations were performed using the lithium acetate method[64].

## Construction of deletion strains
The construction of the *V. parahaemolyticus* RIMD 2210633 surrogate strain and of the *V. parahaemolyticus* BB22OP and *V. coralliilyticus* BAA-450 derivatives was reported previously[30,38]. To construct deletion strains, 1 kb sequences upstream and downstream of each gene to be deleted were cloned into pDM4, a Cm$^R$OriR6K suicide plasmid. The pDM4 constructs were transformed into *E. coli* DH5α (λ-pir) by electroporation, and then transferred into *Vibrio* strains via conjugation. Transconjugants were selected on agar plates supplemented with chloramphenicol, and then counter-selected on agar plates containing 15% (wt/vol) sucrose for loss of the sacB-containing plasmid. Deletions were confirmed by PCR.

## Toxicity assays in *E. coli*
For testing the toxicity of proteins during the growth of bacteria in suspension, *E. coli* strains carrying arabinose-inducible expression plasmids were grown in 2xYT broth supplemented with the appropriate antibiotics and 0.4% (wt/vol) glucose (to repress expression from the P*bad* promotor) at 30 °C. Overnight cultures were washed twice with fresh 2xYT broth, normalized to an OD$_{600}$ of 0.01 in 2xYT broth and then transferred to 96-well plates (200 µl per well) in quadruplicate. Cultures were grown at 37 °C in a BioTek SYNERGY H1 microplate reader with constant shaking (205 cpm). After 2 h, L-arabinose was added to a final concentration of 0.05% (wt/vol) to induce protein expression. OD$_{600}$ readings were acquired every 10 min and data were visualized using GraphPad PRISM v9.

For testing the toxicity of proteins during the growth of bacteria on solid media, *E. coli* strains carrying arabinose-inducible expression plasmids were streaked onto LB agar plates supplemented with the appropriate antibiotics and 0.4% (wt/vol) glucose (repressing plates) or 0.1% (wt/vol) L-arabinose (inducing plates). Plates were incubated for 16 h at 37 °C.

## Protein expression in *E. coli*
*E. coli* cultures harboring arabinose-inducible expression plasmids were grown overnight in 2xYT broth supplemented with appropriate antibiotics to maintain the expression plasmids. Bacterial cultures were then washed with fresh 2xYT and resuspended in 3 ml of 2xYT supplemented with appropriate antibiotics. Next, bacterial cultures were incubated with constant shaking (220 rpm) at 37 °C for 2 h. After 2 h, L-arabinose was added to a final concentration of 0.05% (wt/vol) to induce protein expression, and cultures were grown for 2 additional hours. Cells equivalent to 0.5 OD$_{600}$ units were collected, and cell pellets were resuspended in 50 µl of 2x Tris-glycine SDS sample buffer (Novex, Life Sciences). Next, samples were boiled at 95 °C for 10 min,

and then loaded onto TGX stain-free gels (Bio-Rad) for SDS-PAGE. Proteins were transferred onto nitrocellulose membranes, which were immunoblotted with α-FLAG (Sigma-Aldrich, F1804), α-Myc (Santa Cruz, 9E10, sc-40), or custom-made α-VgrG1[65] antibodies at 1:1000 dilution, as indicated. Finally, protein signals were visualized in a Fusion FX6 imaging system (Vilber Lourmat) using enhanced chemiluminescence (ECL).

### Toxicity assays in yeast
Yeast cells were grown overnight in appropriate SD media[64] supplemented with 4% (wt/vol) glucose, washed twice with ultrapure milli-Q water, and then normalized to an $OD_{600}$ of 1.0 in milli-Q water. Next, 10-fold serial dilutions were spotted onto SD agar plates containing either 4% (wt/vol) glucose (repressing plates) or 2% (wt/vol) galactose and 1% (wt/vol) raffinose (inducing plates). The plates were incubated at 30 °C for two days.

### Protein expression in yeast
Yeast cells harboring galactose-inducible expression plasmids were grown overnight in selective SD media supplemented with glucose, washed twice with ultrapure milli-Q water, and then normalized to $OD_{600} = 1.0$ in selective media supplemented with galactose and raffinose to induce protein expression. Next, the cells were grown at 30 °C for 16 h, and 1.0 $OD_{600}$ units of cells were pelleted and lysed, as previously described[64]. Proteins were detected in immunoblots using α-Myc antibodies at a 1:1000 dilution and α-β-actin antibodies (Santa Cruz, sc-47778, C4) at a 1:500 dilution.

### Bacterial competition assays
Attacker and prey *Vibrio* strains were grown overnight, normalized to an $OD_{600}$ of 0.5, and then mixed at a 4:1 (attacker:prey) ratio in triplicate. Then, 25 μl of the mixtures were spotted onto MLB agar competition plates supplemented with 0.05% (wt/vol) L-arabinose when induction of protein expression from a plasmid was required. Competition plates were incubated at the indicated temperatures for 4 h. The colony-forming units (CFU) of the prey strains at t = 0 h were determined by plating 10-fold serial dilutions on selective media plates. After 4 h of co-incubation of the attacker and prey mixtures on the competition plates, the bacteria were harvested and the CFUs of the surviving prey strains were determined by plating 10-fold serial dilutions on selective media plates. Data were visualized using GraphPad PRISM v9. Statistical tests were performed in Excel 2019.

### Protein secretion assays
*Vibrio* strains were grown overnight in appropriate media and then normalized to an $OD_{600}$ of 0.18 in 3 ml MLB broth supplemented with appropriate antibiotics and 0.01-0.05% (wt/vol) L-arabinose, when expression from an arabinose-inducible plasmid was required. Bacterial cultures were incubated with constant shaking (220 rpm) at the indicated temperatures for the specified durations. For expression fractions (cells), cells equivalent to 0.5 $OD_{600}$ units were collected, and cell pellets were resuspended in 50 μl of 2x Tris-glycine SDS sample buffer (Novex, Life Sciences). For secretion fractions (media), supernatant volumes equivalent to 10 $OD_{600}$ units were filtered (0.22 μm), and proteins were precipitated using the deoxycholate and trichloroacetic acid method[66]. The precipitated proteins were washed twice with cold acetone, and then air-dried before resuspension in 20 μl of 100 mM Tris-Cl (pH = 8.0) and 20 μl of 2x protein sample buffer with 5% β-mercaptoethanol. Next, samples were incubated at 95 °C for 5 or 10 min and then resolved on TGX Stain-free gel (Bio-Rad). The proteins were transferred onto 0.2 μm nitrocellulose membranes using Trans-Blot Turbo Transfer (Bio-Rad) according to the manufacturer's protocol. Membranes were then immunoblotted with α-FLAG (Sigma-Aldrich, F1804), α-Myc (Santa Cruz, 9E10, sc-40), Direct-Blot™ HRP anti-*E. coli* RNA polymerase sigma 70 (mouse mAb #663205;

BioLegend; referred to as α-RNAp) or custom-made α-VgrG1[65] antibodies at 1:1000 dilution. Protein signals were visualized in a Fusion FX6 imaging system (Vilber Lourmat) using ECL reagents.

### Identifying RIX domain-containing proteins
The position-specific scoring matrix (PSSM) of RIX was constructed using the N-terminal 55 residues of Tme1 (WP_015297525.1) from *V. parahaemolyticus* BB22OP. Five iterations of PSI-BLAST were performed against the RefSeq protein database. In each iteration, a maximum of 500 hits with an expect value threshold of $10^{-6}$ were used. Compositional adjustments and filters were turned off. The genomic neighborhoods of RIX domain-containing proteins (Supplementary Data 1) were analyzed as described previously[23,30]. Duplicated protein accessions appearing in the same genome in more than one genomic accession were removed if the same downstream protein existed at the same distance. The T6SS core components in the RIX domain-containing bacterial genomes (Supplementary Data 2) were identified as previously described[16]. Similarity to the T6SS1 cluster proteins of *V. parahaemolyticus* BB22OP (Supplementary Data 2) was evaluated as previously described[30].

### Illustration of the conserved residues of the RIX domain
RIX domain sequences were aligned using Clustal Omega[67]. Aligned columns not found in the RIX domain of Tme1 were discarded. The RIX domain-conserved residues were illustrated using the WebLogo server[68] (https://weblogo.threeplusone.com).

### Constructing a phylogenetic tree of RIX domain-encoding bacterial strains
DNA sequences of *rpoB* were aligned using MAFFT v7.505 FFT-NS-2 (https://mafft.cbrc.jp/alignment/server)[69]. Partial and pseudogene sequences were discarded. The evolutionary history was inferred using the neighbor-joining method[70] with the Jukes-Cantor substitution model (JC69). The analysis included 686 nucleotide sequences and 4,021 conserved sites. The tree was visualized using iTOL (https://itol.embl.de/).

### Protein structure predictions
Predicted protein structures were downloaded from the AlphaFold Protein Structure Database[37,71] in August 2022 (https://alphafold.ebi.ac.uk/). The structures of proteins that were not available on the database and of protein complexes were predicted in ColabFold: AlphaFold2 using MMseqs2[36]. All PDB files used in this work are available as Supplementary Data 3. Protein structures were visualized using ChimeraX v1.4[72].

### Analyses of RIX C-terminal sequences
Amino acid sequences C-terminal to RIX were clustered in two dimensions using CLANS[35]. To predict the activities or domains in each cluster, at least two representative sequences (when more than one was available) were analyzed using the NCBI Conserved Domain Database (CDD)[73] and HHpred (https://toolkit.tuebingen.mpg.de/tools/hhpred)[43]. If no activity or domain could be predicted, the protein sequences were further used for AlphaFold2 structure prediction[36] followed by a 3D protein structure comparison in the Dali server[44]. Predicted antibacterial or anti-eukaryotic activities were determined based on known targets of the C-terminal toxic domains (e.g., an actin cross-linking domain was annotated as anti-eukaryotic, whereas a lysozyme-like domain was annotated as antibacterial) and by the presence or absence of possible immunity gene immediately downstream of the RIX domain-encoding gene (i.e., within 50 base pairs of the RIX domain-encoding gene's stop codon[30]). Notably, for some RIX domain-containing proteins, the activity was experimentally confirmed in this or in past work (e.g., effectors containing Tme and Unknown_2 are confirmed antibacterial effectors).

**Illustration of the conserved residues of Rte1**

Homologs of Rte1 (WP_006958655.1) from *Vibrio coralliilyticus* were identified using PSI-BLAST (4 iterations; a maximum of 500 hits with an expect value threshold of $10^{-6}$ and a query coverage of 70% were used). Rte1 homologs were aligned using Clustal Omega and conserved residues were illustrated using WebLogo 3 (https://weblogo.threeplusone.com/create.cgi).

**Reporting summary**

Further information on research design is available in the Nature Portfolio Reporting Summary linked to this article.

## Data availability

The experimental and computational data that support the findings of this research are available in this article and its supplementary information files. Source data are provided with this paper.

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

## Acknowledgements

This project received funding from the European Research Council under the European Union's Horizon 2020 research and innovation program (grant agreement no. 714224 to DS), and from the Israel Science Foundation (grant no. 920/17 to DS and grant no. 1362/21 to DS and EB). CMF was supported by a scholarship from the Clore Israel Foundation and by a scholarship for outstanding doctoral students from the Orthodox community from the Council for Higher Education. We thank Kinga Keppel and Biswanath Jana for technical assistance, and the rest of the members of the Salomon and Bosis laboratories for helpful discussions and suggestions. This work was performed in partial fulfillment of the requirements for a PhD degree for Katarzyna Kanarek at the Faculty of Medicine, Tel Aviv University.

## Author contributions

K.K.: conceptualization, investigation, methodology, and writing—original draft. C.M.F.: investigation, methodology, and writing—review and editing. E.B.: conceptualization, investigation, methodology, funding acquisition, and writing—original draft. D.S.: conceptualization, supervision, funding acquisition, investigation, methodology, and writing—original draft.

## Competing interests

The authors declare no competing interests.

## Additional information

**Supplementary information** The online version contains
supplementary material available at

Eran Bosis or Dor Salomon.

**Peer review information** *Nature Communications* thanks Patricia Bernal
and the other, anonymous, reviewers for their contribution to the peer
review of this work. A peer review file is available.

