## [Peer Review File · Nature Communications]

The RIX domain defines a class of polymorphic T6SS effectors
and secreted adaptorsReviewer #1 (Remarks to the Author):

In this paper, Kanarek et al. report an arginine-rich sequence in the N-terminus of some T6SS effectors that can be used as a signature domain, termed RIX, for a broad class of effectors in Vibrionaceae. By fusing a heterologous effector Tse1 to a truncated effector Tme1 in *V. parahaemolyticus*, Tse1 appeared to be secreted by the N-terminus Tme1. Using the RIX sequence and bioinformatic analysis, the authors found RIX widespread in Vibrionaceae and identified several effectors in *V. campbellii* and one in *V. coralliilyticus*. Lastly, they demonstrated that a RIX protein in *V. coralliilyticus* was required for the secretion of an antibacterial effector Rte1, whose toxicity was neutralized by its cognate protein Rti1. The authors used protein-structural prediction analysis to suggest that the RIX protein act as a tether for the secretion of Rte1. Overall, the authors have demonstrated the use of the RIX domain for finding new effectors. However, the secretion route of the RIX domain proteins remains unclear. Based on the evidence provided, it appears to be functionally similar to PAAR domain proteins, although the predicted folded structure is different from the cone-shaped PAAR. As a result, despite the large amount of data included in this study, it appears to be weak on the mechanistic side.

Major issues:

- Much of this work relies on the T6SS-dependent secretion assays. However, it seems that the authors did not provide a lysis control nor a positive secretion control for any secretion assay. Therefore, the presence or absence of signals in media could be attributed to cell lysis or inhibition of overall T6SS secretion.
- What is the delivery limit of the RIX domain? Tse1 is a small T6SS effector (~16 kDa). Could larger cargo proteins and non-T6SS proteins be secreted by T6SS through fusing to RIX?
- Fig 5: How could the RIX-containing protein mediate the secretion of downstream Rte1? Apart from bioinformatics analysis, it will be important to also test the direct interaction between RIX and Rte1 by biochemical approaches.
- Line 279-280: the author said "our attempts to determine the T6SS tube-spike component with which RIX domain-containing proteins interact have been inconclusive.". Could RIX interact with other T6SS components or does the interaction between RIX and VgrG or PAAR proteins need additional "chaperone" proteins?

Minor comments:

1. Figure 1B, the blot signal is too weak for the target protein, in contrast to the strong non-specific band.
2. Line 379: there is no result related to the immunoblots of protein expression in yeast
3. Line 400 and 404: different types of two-fold. "2x" and "2X"

Reviewer #2 (Remarks to the Author):

In this manuscript by Kanarek et al., authors report the identification of a new domain, named RIX, that is present in T6SS effectors secreted by members of the Vibrionaceae family. The work is interesting as expands the knowledge about adaptor domains found in T6SS effectors. However, as this is restricted to Vibrionaceae and many other domains with the same function were described before by the same group (MIX, FIX), the work does not bring significant novelty to the field. In addition, the terms chosen by the authors to describe this new domain creates confusion to the field. In my opinion, RIX would be better described as a new adaptor domain that is linked to the C-terminal toxic domains. The concept of adaptors is well established in the T6SS field. It is curious that RIX can also be found fused to other adaptor domains (e.g. TTR domain); however, the concept of adaptors helping the binding of effectors/toxins with some T6SS structural component remains the same.

Major comments about the experimental data, which lacks key control to support hypothesis:

1) Fig 1.: authors mentioned RIX is present in the first 60aa of Tme1, but then decided to test 111 aa (almost double the size) for the fusion with Tse1. Later, 60aa were used for *E. coli* toxicity assays. It is important to be consistent with the sizes in ALL experiments or show the results with both regions.

2) This is a conceptual point: Why authors claim RIX is necessary and "sufficient" for T6SS-

mediated secretion? I agreed it is necessary, but it is for sure not sufficient because it needs other T6SS components for secretion.

3) All experiments shown in Fig 3 need to be repeated with a truncated version of the effectors lacking RIX to actually confirm that the domain is required for secretion.

4) The same problem occurs in Fig. 4. Needs repeating with the effector lacking RIX domain.

5) Line 165: I do not like the term "tether". The field already have the term "adaptor" established for the same role. There is no conceptual novelty here so I believe the same term should be used, but explaining its particularities in Vibrionaceae, which seems to love to mix and match adaptors.

6) Fig 5.: Authors should repeat experiments 5D and 5E deleting RIX and TTR domain independently to confirm whether these are indeed required as hypothesized. As already highlighted by the authors in the discussion (lines 275-276), this mechanism is not new (reference 22).

Minor:

1) Line 48: usually Rhs repeats are present in evolved effectors with PAAR or PARR-like (DUF4150) domains, not in cargo ones.

Reviewer #3 (Remarks to the Author):

In this manuscript, Kanarek et. al., describe a new domain identified in T6SS-related proteins (mainly in putative T6SS effectors and adaptor proteins) that seems to be necessary to load some effectors onto the machinery upon secretion. The domain, located at the N-terminus of the protein, was named RIX (aRginine-rich type sIX). This work and the nomenclature used are a continuation of similar studies performed in this laboratory where the MIX and FIX domains were previously described. The analysis performed in this work follows a pipeline similar to those of the above-mentioned studies. The authors first identified a novel effector with an N-terminal domain of unknown function. They showed that this domain is necessary for T6SS secretion and does not have a toxic effect, which means that it has a structural function in the system, probably to be properly loaded onto the machinery. The authors then identify other proteins that carry this newly identified domain, study the distribution of the domain, and characterize the C-terminal of the RIX-containing protein to understand its implication in T6SS. There is a good correlation between the presence of T6SS clusters and the presence of RIX proteins. Approximately 98% of strains with RIX-containing proteins possess at least one T6SS cluster. Since most RIX proteins have C-terminals with putative toxic domains (anti-bacterial or anti-eukaryotic) or adapter functions, this domain can be used to identify new T6SS effectors. In fact, this is a novelty of this study, where 9 of 502 RIX-containing proteins did not have a C-terminal toxic domain. In these cases, the gene encoding the RIX domain was located upstream of genes encoding putative effector-immunity pairs. This resembles other T6SS-gene architectures where the structure domain (e. g., the PAAR domain) and the effector domain could be encoded in the same gene or adjacent genes. In summary, the present manuscript reveals a comprehensive and well-executed study that identifies a new T6SS domain and novel polymorphic T6SS effectors that are certainly of interest to the field. I have some points that I would like to further discuss with the authors.

Major points:

1. It seems that the RIX domain is important for protein stability in the cytosol (Fig. 1B). Is this a general characteristic of the different RIX-containing proteins? Could it be one of the functions of the domain? If the absence of the RIX domain prevents the binding of the effector to a structural component of the system, it might explain that an effector lacking RIX is unstable in the cytosol, similar to effectors chaperoned by Hcp (Tse2 among others).

2. There appears to be residual T6SS-independent secretion of Tme1 that surprisingly also appears when expressing Tme161-310. This is not expected since this strain does not express the complete

Tme1 protein. Could the authors discuss what this could be? Is this band unspecific? Why is it present only in the Media fraction and not in the Cells fraction? Could this band be the one that the FLAG antibody unspecifically detects in Fig. 1C? This is all very confusing, and the authors should make an effort to make this point crystal clear. An unspecific band of the size of the band to be studied is not ideal and limits the capacity of the authors to draw unambiguous conclusions.

3. The expression of Tse1 fused to the RIX domain is very poor compared to the expression of the original protein. Could the authors discuss this point? It is difficult to see expression and secretion in this figure; could it be improved? The authors said that they have to use the first 111 amino acids for the fused protein to be stable, whereas the RIX-domain seems to be shorter (65 amino acids), would this depend on the C-terminal fused effector? Have they tried with any other effector? Tse1 is known to be chaperoned by Hcp; the presence of N-terminal RIX might prevent this interaction, and thus a different effector might be more permissive to an N-terminal fusion and would allow more clear results for this statement.

4. It is curious to see that the RIX domain is present only in the Vibrionaceae family. How could that be explained as T6SS being transferred mainly by HGT. What could explain this specialization? It might be related to the T6SS interaction partner of this domain, which is the missing part of this manuscript. The authors should add a column to the Supplementary DataSet S2 to include the phylogeny of the T6SSs present in the strains with RIX-proteins. This could clarify the nature or origin of the RIX-containing proteins. For example, if strains with RIX proteins all harbour T6SS from phylogenetic group 1 (group i), it could be because RIX domains interact with a specific structural component (or domain) only present in this phylogenetic group of T6SSs that has diverged in the Vibrionaceae family.

5. The fact that genes encoding RIX-containing proteins are only found as orphan genes and never within T6SS clusters is also very puzzling. Is it never identified next to any T6SS gene such as orphans hcp or vgrG genes? A more comprehensive discussion regarding the genomic context of gene encoding RIX proteins would help elucidate the specific presence in Vibrionaceae and, more importantly, could reveal the T6SS interaction partner(s).

6. The authors described in the discussion that studies to identify the structural component(s) of T6SS that interact with RIX proteins are inconclusive. The authors should show the approaches they have used and include the data as supplementary information. In this way, reviewers could provide their input on this matter, which is the main weakness of this manuscript. To characterize the RIX domain as related to T6SS, it is key to show that it interacts with a structural component of T6SS. RIX proteins are expected to interact with the delivery part of the system (PAAR, VgrG, or Hcp proteins). Have they made any predictions with Alphafold or any biochemistry assay?

7. In Table 1, the authors classified the RIX-associated C-terminal extensions and divided them into their predicted roles. Some activity/domains appear as unknown or DUF; how do they know that they belong to the antibacterial or the anti-eukaryotic group? Is it due only to the presence/absence of the putative immunity protein? How do the authors determine the presence/absence of the putative immunity protein? (Lanes 11-114) There are no data on this statement.

8. It is very curious that approximately half of the RIX-containing effectors are anti-eukaryotic effectors, while the general proportion of anti-eukaryotic T6SS effectors is much smaller. The authors should refer to this in the discussion and give some possibilities that could explain this. Could it be because most strains with RIX-containing proteins are pathogenic? Have they done any analysis on these?

Minor comments:

1. The authors might consider editing 'Toxin domain' expression and using "toxic domain" (lines 18, 39, 49, ...).

2. The authors refer to 'anti-eukaryotic T6SSs'. This is not an appropriate classification, since the same system can deliver both types of effectors, as described for *P. aeruginosa* and *Vibrio*. It

would be better to refer to systems that deliver only antibacterial effectors and systems that deliver both types of effectors.

3. The authors use the word 'missile' to refer to the tail structure of the T6SS. It is a good analogy, but it is used multiple times in the same paragraph (lines 34-45) which sounds a bit repetitive.

4. Supplementary Figure S1- Please edit the name of the protein in the third lane instead of Tme161-110, it should say Tme161-310.

A point-by-point response to reviewers' comments

Reviewer #1 (Remarks to the Author):

In this paper, Kanarek et al. report an arginine-rich sequence in the N-terminus of some T6SS effectors that can be used as a signature domain, termed RIX, for a broad class of effectors in Vibrionaceae. By fusing a heterologous effector Tse1 to a truncated effector Tme1 in *V. parahaemolyticus*, Tse1 appeared to be secreted by the N-terminus Tme1. Using the RIX sequence and bioinformatic analysis, the authors found RIX widespread in Vibrionaceae and identified several effectors in *V. campbellii* and one in *V. coralliilyticus*. Lastly, they demonstrated that a RIX protein in *V. coralliilyticus* was required for the secretion of an antibacterial effector Rte1, whose toxicity was neutralized by its cognate protein Rti1. The authors used protein-structural prediction analysis to suggest that the RIX protein act as a tether for the secretion of Rte1. Overall, the authors have demonstrated the use of the RIX domain for finding new effectors. However, the secretion route of the RIX domain proteins remains unclear. Based on the evidence provided, it appears to be functionally similar to PAAR domain proteins, although the predicted folded structure is different from the cone-shaped PAAR. As a result, despite the large amount of data included in this study, it appears to be weak on the mechanistic side.

Major issues:

- Much of this work relies on the T6SS-dependent secretion assays. However, it seems that the authors did not provide a lysis control nor a positive secretion control for any secretion assay. Therefore, the presence or absence of signals in media could be attributed to cell lysis or inhibition of overall T6SS secretion.

Following this comment, we now include VgrG1 (positive T6SS secretion control) and RNA polymerase sigma 70 (loading and lysis control) detection in all secretion assays. The results confirm that the signals detected in the media are not attributed to cell lysis and that their absence is not due to inhibition of overall T6SS activity. Please see the revised Fig. 1B, 1C, 3B, 4A, 4B, 5D, and Supplementary Fig. S7A.

- What is the delivery limit of the RIX domain? Tse1 is a small T6SS effector (~16 kDa). Could larger cargo proteins and non-T6SS proteins be secreted by T6SS through fusing to RIX?

We now show that RIX (the first 60 amino acids of Tme1) is sufficient to drive the T6SS-mediated secretion of sfGFP, a well-folded, non-T6SS-related protein (see the revised Fig. 1C). Although we did not directly determine the delivery limit of RIX, we showed the T6SS-mediated secretion of the 526 amino acid-long (~60 kDa) RIX-containing effector WP_005530005.1 (see Fig. 3B). Notably, the longest

RIX-containing protein identified in our analyses is 549 amino acid-long (WP_192890785.1).

• Fig 5: How could the RIX-containing protein mediate the secretion of downstream Rte1? Apart from bioinformatics analysis, it will be important to also test the direct interaction between RIX and Rte1 by biochemical approaches.

We tested the interaction between the RIX-containing tether and Rte1, including various truncated forms, in co-IP and pull-down experiments. Unfortunately, as with our attempt to biochemically identify a T6SS structural binding partner for RIX-containing proteins (see below), the results were inconclusive. Therefore, we are unable to experimentally confirm that the two proteins physically interact via the domains proposed by the AlphaFold2 prediction at this time. Nevertheless, we genetically show the requirement of both the RIX and the TTR domains for Rte1 T6SS-mediate secretion (see revised Fig. 5E). Taken together with the high-confidence structure prediction and the known role of TTR domains in mediating protein-protein interactions, the results strongly support the proposed role of this RIX-containing protein as a tether for Rte1.

• Line 279-280: the author said “our attempts to determine the T6SS tube-spike component with which RIX domain-containing proteins interact have been inconclusive.”. Could RIX interact with other T6SS components or does the interaction between RIX and VgrG or PAAR proteins need additional “chaperone” proteins?

As mentioned in the original version of the manuscript, we investigated the possible interaction of RIX proteins with secreted tube-spike components (Hcp, VgrG, and PAAR) in various co-IP and pull-down assays under diverse conditions, without reaching conclusive results. Following the reviewers’ comments, we made additional attempts with various other tags and antibodies, as well as split-reporter assays. Still, the results are inconclusive and we are unable to determine the interacting partner of RIX at this time. We acknowledge the possibility that there might be another protein required to mediate this interaction; however, since no candidate is known at this time and since no protein that could act as such an adaptor is commonly encoded next to RIX-containing proteins, this remains out of scope for this manuscript.

We speculate that because RIX was able to deliver sfGFP via the T6SS (which is considered a well-folded protein and is, therefore, unlikely to fit inside the tube), the protein is not loaded inside the Hcp tube. However, without additional evidence indicating loading onto a specific spike component, we cannot determine this with certainty. We elaborate on this further in the Discussion section (lines 300-310).

Minor comments:

1. Figure 1B, the blot signal is too weak for the target protein, in contrast to the strong non-specific band.

We provide a new experiment for Fig. 1B (with a different tag and detecting antibody) that clearly shows the RIX-dependent secretion of Tme1 without an overlapping non-specific band.

2. Line 379: there is no result related to the immunoblots of protein expression in yeast

We thank the reviewer for pointing this out. We added Supplementary Fig. S6, showing the expression of the three predicted anti-eukaryotic RIX-containing proteins in yeast. We could detect the expression of the two predicted deamidase effectors in yeast. Although we could not detect the expression of the effector with the actin cross-linking domain (possibly due to the fact that it is very toxic, even under repressing conditions), we showed that its expression induced actin cross-linking in yeast cells. Please see also the accompanying text (lines 154-156).

3. Line 400 and 404: different types of two-fold. “2x” and “2X”

Thank you for pointing this out. We proofread the text and corrected such inconsistencies.

Reviewer #2 (Remarks to the Author):

In this manuscript by Kanarek et al., authors report the identification of a new domain, named RIX, that is present in T6SS effectors secreted by members of the Vibrionaceae family. The work is interesting as it expands the knowledge about adaptor domains found in T6SS effectors. However, as this is restricted to Vibrionaceae and many other domains with the same function were described before by the same group (MIX, FIX), the work does not bring significant novelty to the field.

We would like to emphasize what we see as the novelty of this work, which we tried to clarify in the revised version of the manuscript:

- 1) We show that RIX is necessary and sufficient for T6SS delivery. Notably, the only two previously described T6SS-specific “marker domains”, MIX and FIX, have not yet been shown as sufficient for T6SS-mediated delivery.**
- 2) Unlike MIX and FIX domains, which are often encoded within T6SS clusters or auxiliary operons, RIX-containing proteins are orphans. Therefore, the identification of these proteins as T6SS effectors would have been challenging without the discovery of RIX, since they lack a clear genetic association with T6SS genes.**
- 3) The fact that RIX is restricted to *Vibrionaceae* reveals a new concept of family-specific delivery domains, which was not previously reported.**
- 4) RIX-containing proteins are often fused to C-terminal toxic domains that are predicted to target eukaryotes, thereby significantly enlarging the repertoire of known T6SS anti-eukaryotic effectors, and indicating that the community did not fully appreciate the potential role of *Vibrio* T6SSs in *Vibrio*-eukaryotes interactions. MIX and FIX-containing effectors are predominantly predicted to be antibacterial.**

In addition, the terms chosen by the authors to describe this new domain creates confusion to the field. In my opinion, RIX would be better described as a new adaptor domain that is linked to the C-terminal toxic domains. The concept of adaptors is well established in the T6SS field. It is curious that RIX can also be found fused to other adaptor domains (e.g. TTR domain); however, the concept of adaptors helping the binding of effectors/toxins with some T6SS structural component remains the same.

We understand the reviewer’s point of view. However, we wish to point out that according to the widely accepted working model described by Unterweger et al., 2017 (PMID: 27856117), effectors are exported through the T6SS whereas adaptor proteins, such as DUF4123, DUF1795, and DUF2169, remain inside the cell. In our case, the RIX domain-containing proteins are secreted. Furthermore, even when RIX-containing proteins serve as tethers, they are secreted independently of their cognate cargo effector. For this reason, we chose not to describe the RIX domain as a T6SS adaptor. This is discussed further in response to the reviewer’s

comment #5.

Major comments about the experimental data, which lacks key control to support hypothesis:

1) Fig 1.: authors mentioned RIX is present in the first 60aa of Tme1, but then decided to test 111 aa (almost double the size) for the fusion with Tse1. Later, 60aa were used for E. coli toxicity assays. It is important to be consistent with the sizes in ALL experiments or show the results with both regions.

We thank the reviewer for pointing this out. We now show that the first 60 aa of Tme1, which were required for Tme1 T6SS-mediated secretion, are sufficient to drive the secretion of the non-T6SS protein sfGFP via the T6SS (See revised Fig. 1C).

2) This is a conceptual point: Why authors claim RIX is necessary and "sufficient" for T6SS-mediated secretion? I agreed it is necessary, but it is for sure not sufficient because it needs other T6SS components for secretion.

We thank the reviewer for raising this point. Please note that we do not challenge the notion that the T6SS itself is required for secretion of a RIX-containing protein. Our claims regarding RIX's role in secretion are only relevant in the context of a functional T6SS, as indicated by the term "T6SS-mediated secretion". We now clarify this issue using sfGFP, which is not a "natural" T6SS-delivered protein, yet it is secreted by the T6SS when fused to RIX (please see the revised Fig. 1C). We also amended the Abstract text to reflect that.

3) All experiments shown in Fig 3 need to be repeated with a truncated version of the effectors lacking RIX to actually confirm that the domain is required for secretion.

We now include truncated versions of the three RIX-containing proteins used in the secretion assays and confirm that their RIX domain is necessary for T6SS-mediated secretion (please see the revised Fig. 3B).

4) The same problem occurs in Fig. 4. Needs repeating with the effector lacking RIX domain.

Please see our response to comment #3. We now experimentally show that the RIX domain is indispensable for the secretion of four different effectors (i.e., Tme1 in Fig. 1B and the three effectors in Fig. 3B). Therefore, we believe that our conclusion that RIX is required for T6SS-mediated secretion is well substantiated even without experimental demonstration for all the proteins discussed in this work.

5) Line 165: I do not like the term "tether". The field already have the term "adaptor" established for the same role. There is no conceptual novelty here so I believe the same term should be used, but explaining its particularities in Vibrionaceae, which seems to love to mix and match adaptors.

We suggest that RIX is found within proteins that are either effectors, if they have a C-terminal toxic domain, or tethers, if they have a protein-protein binding domain. Meaning, not all RIX domain-containing proteins are tethers. Moreover, as noted above, the term “adaptor” has been used in the field to describe a set of proteins that are not secreted together with their cognate effector via the T6SS (e.g., DU4123, DUF2169, and DUF1795). Since RIX-containing effectors are secreted via T6SS with their RIX domain and RIX-containing tethers are secreted via T6SS even in the absence of their cognate cargo effector (a previously-undescribed mechanism), we chose not to define RIX as a T6SS adaptor.

6) Fig 5.: Authors should repeat experiments 5D and 5E deleting RIX and TTR domain independently to confirm whether these are indeed required as hypothesized. As already highlighted by the authors in the discussion (lines 275-276), this mechanism is not new (reference 22).

Following the reviewer’s suggestion, we tested additional constructs in which the RIX-containing protein is truncated at the N- or C-terminus, and we show that both domains are required to deliver Rte1 into prey cells during interbacterial competitions (please see the revised Fig. 5E).

We would also like to emphasize that the proposed secretion mechanism is new. We only mention that in some aspects it resembles the previously reported “binary effector module” mechanism described in reference #22 (i.e., the secretion of a cargo effector with the help of a MIX domain-containing co-effector). The two mechanisms are distinct, since the co-effector in the “binary module” is not secreted independently of the effector, whereas the RIX-containing protein is secreted via T6SS independently of the cargo effector.

Minor:

1) Line 48: usually Rhs repeats are present in evolved effectors with PAAR or PARR-like (DUF4150) domains, not in cargo ones.

We thank the reviewer for pointing this out. We now mention the fact that many Rhs-containing effectors are specialized effectors fused to PAAR, in lines 52-53.

Reviewer #3 (Remarks to the Author):

In this manuscript, Kanarek et. al., describe a new domain identified in T6SS-related proteins (mainly in putative T6SS effectors and adaptor proteins) that seems to be necessary to load some effectors onto the machinery upon secretion. The domain, located at the N-terminus of the protein, was named RIX (aRginine-rich type sIX). This work and the nomenclature used are a continuation of similar studies performed in this laboratory where the MIX and FIX domains were previously described. The analysis performed in this work follows a pipeline similar to those of the above-mentioned studies. The authors first identified a novel effector with an N-terminal domain of unknown function. They showed that this domain is necessary for T6SS secretion and does not have a toxic effect, which means that it has a structural function in the system, probably to be properly loaded onto the machinery. The authors then identify other proteins that carry this newly identified domain, study the distribution of the domain, and characterize the C-terminal of the RIX-containing protein to understand its implication in T6SS. There is a good correlation between the presence of T6SS clusters and the presence of RIX proteins. Approximately 98% of strains with RIX-containing proteins possess at least one T6SS cluster. Since most RIX proteins have C-terminals with putative toxic domains (anti-bacterial or anti-eukaryotic) or adapter functions, this domain can be used to identify new T6SS effectors. In fact, this is a novelty of this study, where 9 of 502 RIX-containing proteins did not have a C-terminal toxic domain. In these cases, the gene encoding the RIX domain was located upstream of genes encoding putative effector-immunity pairs. This resembles other T6SS-gene architectures where the structure domain (e. g., the PAAR domain) and the effector domain could be encoded in the same gene or adjacent genes. In summary, the present manuscript reveals a comprehensive and well-executed study that identifies a new T6SS domain and novel polymorphic T6SS effectors that are certainly of interest to the field. I have some points that I would like to further discuss with the authors.

We thank the reviewer for the kind words and for the helpful comments.

Major points:

1. It seems that the RIX domain is important for protein stability in the cytosol (Fig. 1B). Is this a general characteristic of the different RIX-containing proteins? Could it be one of the functions of the domain? If the absence of the RIX domain prevents the binding of the effector to a structural component of the system, it might explain that an effector lacking RIX is unstable in the cytosol, similar to effectors chaperoned by Hcp (Tse2 among others).

The observation that truncation of the RIX domain affects protein stability is inconsistent. In Tme1 (see the revised Fig. 1B), removing RIX resulted in a higher apparent expression level of the protein. However, truncating RIX from two of the three effectors investigated in the revised Fig. 3B resulted in apparent lower expression levels. Given these varying phenotypes and the fact that these

proteins are over-expressed from a plasmid, we cannot conclude that RIX plays a role in protein stability or that a structural T6SS component serves as a chaperone for RIX.

2. There appears to be residual T6SS-independent secretion of Tme1 that surprisingly also appears when expressing Tme161-310. This is not expected since this strain does not express the complete Tme1 protein. Could the authors discuss what this could be? Is this band unspecific? Why is it present only in the Media fraction and not in the Cells fraction? Could this band be the one that the FLAG antibody unspecifically detects in Fig. 1C? This is all very confusing, and the authors should make an effort to make this point crystal clear. An unspecific band of the size of the band to be studied is not ideal and limits the capacity of the authors to draw unambiguous conclusions.

We thank the reviewer for this comment. Indeed, the observed band that appeared at the same size as the full-length Tme1 is a non-specific band that was also observed in the absence of a full-length Tme1 and in the supernatant fractions of the original Fig. 1C. We agree that this is confusing, and therefore we repeated this experiment with a different C-terminal tag fused to Tme1 (Myc instead of FLAG). As shown in the revised Fig. 1B, now there is no unspecific band corresponding to the same size as the full-length Tme1. Therefore, our conclusion that RIX is necessary for the T6SS-mediated secretion of Tme1 stands true.

3. The expression of Tse1 fused to the RIX domain is very poor compared to the expression of the original protein. Could the authors discuss this point? It is difficult to see expression and secretion in this figure; could it be improved? The authors said that they have to use the first 111 amino acids for the fused protein to be stable, whereas the RIX-domain seems to be shorter (65 amino acids), would this depend on the C-terminal fused effector? Have they tried with any other effector? Tse1 is known to be chaperoned by Hcp; the presence of N-terminal RIX might prevent this interaction, and thus a different effector might be more permissive to an N-terminal fusion and would allow more clear results for this statement.

Following this comment and a comment made by reviewer #2, we tested additional RIX fusions. We now provide a revised Fig. 1C in which we clearly show that RIX (i.e., the first 60 aa of Tme1) is sufficient to drive the secretion of a stable sfGFP via the T6SS.

4. It is curious to see that the RIX domain is present only in the Vibronaceae family. How could that be explained as T6SS being transferred mainly by HGT. What could explain this specialization? It might be related to the T6SS interaction partner of this domain, which is the missing part of this manuscript. The authors should add a column to the Supplementary DataSet S2 to include the phylogeny of the T6SSs present in the strains with RIX-proteins. This could clarify the nature or origin of the RIX-containing

proteins. For example, if strains with RIX proteins all harbour T6SS from phylogenetic group 1 (group i), it could be because RIX domains interact with a specific structural component (or domain) only present in this phylogenetic group of T6SSs that has diverged in the Vibrionaceae family.

To address the reviewer's comment, we first determined whether RIX proteins are associated with a specific T6SS in vibrios. We used the components of T6SS1 from *V. parahaemolyticus* strain BB22OP (belonging to phylogenetic group i, according to Boyer et al, 2009), which is the system through which Tme1 is secreted, and asked whether a similar system is found in all strains encoding RIX proteins. As seen in the revised Supplementary Dataset S2, a *V. parahaemolyticus* T6SS1-like cluster is not found in ~15% of the RIX-containing strains. Moreover, some of these strains only have a T6SS belonging to a different phylogenetic group (e.g., *V. parahaemolyticus* MAVP56, which only has T6SS2 belonging to phylogenetic group v, according to Boyer et al). Therefore, we do not find a clear connection between the presence of RIX and a specific T6SS that can hint at a possible interacting structural component of the T6SS. Notably, we cannot say whether RIX-containing proteins found in strains lacking a group i T6SS are actively secreted by the T6SS encoded by the relevant genome or not. This is now described in the text (lines 105-110). As for the presence of RIX predominantly in *Vibrionaceae*, we can only speculate that either it is found on mobile elements that are horizontally shared specifically between members of this family, or that RIX-containing proteins are secreted by T6SSs that are restricted to this bacterial family and are therefore only maintained by its members. We now elaborate on this further in the Discussion section (lines 279-289).

5. The fact that genes encoding RIX-containing proteins are only found as orphan genes and never within T6SS clusters is also very puzzling. Is it never identified next to any T6SS gene such as orphans hcp or vgrG genes? A more comprehensive discussion regarding the genomic context of gene encoding RIX proteins would help elucidate the specific presence in Vibrionaceae and, more importantly, could reveal the T6SS interaction partner(s).

The data showing the genomic neighborhood (10 genes upstream and 10 genes downstream) of all RIX-containing proteins are found in Supplementary Dataset S1. These data show that there are a few cases in which an Hcp or a PAAR/PAAR-like-encoding gene is found in the RIX neighborhood; however, these are not found in the same operon as RIX, and they are separated from RIX by non-T6SS genes. There are 2 very similar yet unique protein accession numbers (WP_240696908.1 and WP_241908865.1) in which RIX is actually fused to an N-terminal PAAR domain. We failed to mention this in the previous version of the manuscript, and it is now mentioned in lines 108-110. Nevertheless, these are the only direct genomic link between RIX and the T6SS. Notably, this PAAR domain is not related to the PAAR-like domain belonging to the T6SS we have shown to be

the one secreting RIX in *V. parahaemolyticus* and *V. coralliilyticus*. Therefore, we are unable to use a “guilt by association” strategy based on RIX neighborhoods to hypothesize which tube-spike component of the T6SS is the loading platform for RIX proteins, and whether or not there is another component or adaptor required for RIX loading onto the tube-spike.

6. The authors described in the discussion that studies to identify the structural component(s) of T6SS that interact with RIX proteins are inconclusive. The authors should show the approaches they have used and include the data as supplementary information. In this way, reviewers could provide their input on this matter, which is the main weakness of this manuscript. To characterize the RIX domain as related to T6SS, it is key to show that it interacts with a structural component of T6SS. RIX proteins are expected to interact with the delivery part of the system (PAAR, VgrG, or Hcp proteins). Have they made any predictions with AlphaFold or any biochemistry assay?

As mentioned in the original version of the manuscript, we investigated the possible interaction of RIX proteins with secreted tube-spike components (Hcp, VgrG, and PAAR) in various co-IP and pull-down assays under diverse conditions, without reaching conclusive results. Following the reviewers’ comments, we made additional attempts with various other tags and antibodies, as well as split-reporter assays. Still, results are either negative or inconsistent. Therefore, without a conclusive result that we are confident in, we are unable to determine the interacting partner of RIX at this time. We do not think it is appropriate to provide dozens of immunoblots with varying results to the readers or the reviewers, since we do not intend to publish a result that we are unsure of and that we cannot confidently stand behind or even reliably interpret. It is possible that a yet-unknown component is required to mediate the loading of RIX onto a tube-spike component, but at this time we have no candidates to test since the genomic analyses described above did not reveal one. AlphaFold2 predictions using various combinations of RIX-containing proteins and T6SS components yielded no confident interaction. We speculate that because RIX is able to deliver sfGFP via the T6SS, the protein is not loaded inside the Hcp tube (since sfGFP is folded and bulky); however, without direct evidence indicating loading on a specific spike component, we cannot determine this with certainty. We now elaborate on these points further in the text (lines 300-310). We wish to note that even in the absence of the direct interacting partner of RIX, the multiple results showing that RIX is necessary and sufficient for T6SS-mediated secretion unequivocally link RIX to the T6SS and demonstrate that RIX is a T6SS delivery domain.

7. In Table 1, the authors classified the RIX-associated C-terminal extensions and divided them into their predicted roles. Some activity/domains appear as unknown or

DUF; how do they know that they belong to the antibacterial or the antieukaryotic group? Is it due only to the presence/absence of the putative immunity protein? How do the authors determine the presence/absence of the putative immunity protein? (Lanes 11-114) There are no data on this statement.

We determined the putative activity based on the presence or absence of a possible or known immunity gene immediately downstream of a RIX-encoding gene (a relevant lane is now added to Table 1) and based on the identity of the C-terminal domains (when known). A possible downstream immunity gene was identified as a gene starting less than 50 nucleotides downstream of the RIX gene stop codon on the same strand. This was revealed by the information available in Supplementary Dataset S1 and by manual validation. These details are now mentioned in Table 1 footnotes and elaborated in the relevant Methods section (lines 476-483). Please also note that the antibacterial activity of a RIX-containing effector with a C-terminal Unknown_2 domain was experimentally validated (Fig. 3A).

8. It is very curious that approximately half of the RIX-containing effectors are antieukaryotic effectors, while the general proportion of antieukaryotic T6SS effectors is much smaller. The authors should refer to this in the discussion and give some possibilities that could explain this. Could it be because most strains with RIX-containing proteins are pathogenic? Have they done any analysis on these?

This point is indeed one of the main novel findings in this work. We hypothesize that these anti-eukaryotic effectors are not necessarily acting to mediate the colonization of an animal host (i.e., pathogenicity), but rather that they act in defense against predation by protists, which are ubiquitous in the marine environment (similar to the observed role of the *V. cholerae* T6SS in Pukatzki et al, 2006). We now elaborate on this in the Discussion section (lines 273-278).

Minor comments:

1. The authors might consider editing 'Toxin domain' expression and using "toxic domain" (lines 18, 39, 49, ...).

The text was edited as suggested.

2. The authors refer to 'antieukaryotic T6SSs'. This is not an appropriate classification, since the same system can deliver both types of effectors, as described for *P. aeruginosa* and *Vibrio*. It would be better to refer to systems that deliver only antibacterial effectors and systems that deliver both types of effectors.

The relevant text was amended to clarify that we mean anti-eukaryotic effectors (Lines 275-278).

3. The authors use the word 'missile' to refer to the tail structure of the T6SS. It is a good analogy, but it is used multiple times in the same paragraph (lines 34-45) which sounds a bit repetitive.

We changed one of the three occurrences of the word “missile” to “ejected apparatus” (line 35).

4. Supplementary Figure S1- Please edit the name of the protein in the third lane instead of Tme161-110, it should say Tme161-310.

Thank you for pointing this out. We revised the Figure accordingly.

Reviewer #1 (Remarks to the Author):

In the revised manuscript, the authors have successfully addressed some of my previous comments. However, they have not adequately addressed the crucial aspect of how the T6SS recognizes and secretes the RIX-containing proteins. Although I agree that some of the novelty lies in the discovery of RIX-domain secreted proteins, claiming its novel function as a secreted tether still needs more experimental evidence to demonstrate its secretion mechanism. They could try cross-linking mass spectrometry or test deletion mutants of *vgrG* or *PAAR* genes, for example. Interestingly, Fig 3b shows variable levels of VgrG1 secretion when different RIX proteins were expressed, suggesting a possible link between RIX and VgrG1. I also agree with comments from other reviewers that the term "tether" might be confusing to the field since it remains unclear what these RIX domain proteins tether to for secretion.

Minor comment: The y-axis title of Figure S8D needs to be fixed.

Reviewer #2 (Remarks to the Author):

The technical problems of the manuscript have been resolved in the new version; however, in my opinion the manuscript is still overall incremental to the current knowledge of the T6SS field as it stands. I think a great contribution that would make this work complete and worth publishing at Nat Comm would have been the mechanistic details of how RIX is interacting with the T6SS machinery.

Reviewer #3 (Remarks to the Author):

The authors have done an excellent job in reviewing the manuscript, putting a tremendous effort into this revision. I am highly satisfied with their responses to the concerns and suggestions raised during the review process. I have no further comments to add but to congratulate the authors for their beautiful work.

A point-by-point response to reviewers' comments

Reviewer #1 (Remarks to the Author):

In the revised manuscript, the authors have successfully addressed some of my previous comments. However, they have not adequately addressed the crucial aspect of how the T6SS recognizes and secretes the RIX-containing proteins. Although I agree that some of the novelty lies in the discovery of RIX-domain secreted proteins, claiming its novel function as a secreted tether still needs more experimental evidence to demonstrate its secretion mechanism. They could try cross-linking mass spectrometry or test deletion mutants of *vgrG* or *PAAR* genes, for example. Interestingly, Fig 3b shows variable levels of VgrG1 secretion when different RIX proteins were expressed, suggesting a possible link between RIX and VgrG1.

We agree with the reviewer that identifying the mechanism by which RIX-containing proteins are loaded onto the T6SS tube-spike is important and interesting. Since we suspect that a yet unidentified component plays a role in this process, we cannot address this in the current work. We will continue investigating this mechanism and report any significant findings in a future manuscript.

I also agree with comments from other reviewers that the term "tether" might be confusing to the field since it remains unclear what these RIX domain proteins tether to for secretion.

We re-define the relevant, non-toxic RIX-containing proteins as “secreted adaptors” rather than “tethers” in the revised manuscript, to conform with the reviewers’ suggestions and concerns.

Minor comment: The y-axis title of Figure S8D needs to be fixed.

The panel was corrected.

Reviewer #2 (Remarks to the Author):

The technical problems of the manuscript have been resolved in the new version; however, in my opinion the manuscript is still overall incremental to the current knowledge of the T6SS field as it stands. I think a great contribution that would make this work complete and worth publishing at Nat Comm would have been the mechanistic details of how RIX is interacting with the T6SS machinery.

We agree with the reviewer that identifying the mechanism by which RIX-containing proteins are loaded onto the T6SS tube-spike is important and

interesting. Since we suspect that a yet unidentified component plays a role in this process, we cannot address this in the current work. We will continue investigating this mechanism and report any significant findings in a future manuscript.

Reviewer #3 (Remarks to the Author):

The authors have done an excellent job in reviewing the manuscript, putting a tremendous effort into this revision. I am highly satisfied with their responses to the concerns and suggestions raised during the review process. I have no further comments to add but to congratulate the authors for their beautiful work.

We thank the reviewer for the valuable discussion and suggestions.